# Observation of hyperbolic intersubband polaritons in native-dielectric-doped van der Waals semiconductor quantum wells

Yue Luo [1,2,3] ✉, Dapeng Ding[3,4], Andres M. Mier Valdivia[3], Daniel T. Larson [3], Song Liu[5], Hong Kuan Ng[3], Jing Wu [1], Kenji Watanabe [6], Takashi Taniguchi [7], Efthimios Kaxiras[3], Hongkun Park [4], Philip Kim [3] & William L. Wilson [2] ✉

Highly doped semiconductor quantum wells (QWs) exhibit strong intersubband transitions resulting from nanoscale electron confinement. Coupling photons to these collective dipoles in this anisotropic quantum structure enables intersubband polaritons with strong nonlinear optical response and hyperbolicity. Analogous to epitaxially grown multi-quantum wells, two-dimensional (2D) van der Waals (vdW) semiconductor heterostructures provide a compelling alternative platform, offering additional degrees of freedom and exceptional optoelectronic properties. Here we report intersubband polaritons in multilayer vdW $WSe_2$ QWs with broadband tunability. By oxidizing the top $WSe_2$ layer into a self-limiting native oxide, we activate charge transfer–induced efficient, high-density doping, enabling strong intersubband transitions and directly visualized polariton propagation. Lithographically defined nanostructures reveal their hyperbolic nature and sub-diffractional confinement, while electrostatic gating offers dynamic dispersion control. These results position vdW multilayers as a highly adaptable platform for tunable mid-infrared nanophotonics and integrated polaritonic circuits, detectors, and light sources.

The quantum confinement of charge carriers in low-dimensional materials leads to emergence of quantized energy states, mapped onto the edges of the systems conduction and valence bands. Intersubband transitions represent a particular set of absorption or emission process, whereby electrons (or holes) in a doped semiconductor undergo transitions between distinct subbands confined within the conduction or valence bands[1,2]. These transitions are observed and exploited in semiconductor quantum wells (QWs) where one-dimensional constraint is imposed on their free carrier motion[3–5]. The harnessing of the intersubband transitions oscillator strength has been crucial for the conceptualization and fabrication of an array of optoelectronic

devices, including quantum cascade lasers[6] (QCLs) and infrared photodetectors[7]. Strong coupling and ultra-strong coupling between cavity photons and individual intersubband transitions have been demonstrated as cavity intersubband polaritons (ISPs) using a variety of different photonic cavity structures containing multi-QWs with high carrier density[8–10]. By engineering the design of the QW semiconductor heterostructures, both transition energies and dipole moments between the subbands can be tailored. Consequently, these ISPs are electrically tunable through the terahertz and mid-infrared spectrum range, with collective properties, such as hyperbolicity[11] and nonlinearity[12]. A wide array of device applications have been

[1]School of Electronic Science and Engineering, Southeast University, Nanjing, Jiangsu, China. [2]Center for Nanoscale Systems, Harvard University, Cambridge, MA, USA. [3]Department of Physics, Harvard University, Cambridge, MA, USA. [4]Department of Chemistry and Chemical Biology, Harvard University, Cambridge, MA, USA. [5]Institute of Microelectronics, Chinese Academy of Sciences, Beijing, China. [6]International Center for Materials Nanoarchitectonics, National Institute for Materials Science, Ibaraki, Japan. [7]Research Center for Functional Materials Science, National Institute for Materials Science, Ibaraki, Japan. ✉e-mail: yueluo@seu.edu.cn; wwilson@cns.fas.harvard.edu

demonstrated such as realizing unconventional light sources[13,14], inversionless polariton laser[15], ultrafast terahertz saturable absorber[16] and optical switches[17].

Emerging 2D van der Waals materials, such as transition metal dichalcogenides (TMDs), are an alternative platform that form natural QWs where charge carriers are quantum confined within the atomically thin layers[18–20]. In addition, the energy separation between the quantized states in few-layer TMDs is governed by the interlayer coupling and can be controlled via engineering the multi-layer structure, geometry, and composition. For instance, the valence band and conduction band splitting in few layers of tungsten diselenide (WSe$_2$) are about few hundred meV at $\Gamma$ and $K$ valley[18,21,22], where intersubband transitions are within the mid-infrared to few THz energy range. These TMD layers with atomically smooth interfaces can be further assembled into heterostructures with other types of TMDs, leading to controlled band realignment and an extended range of intersubband transition energies[23]. Additionally, forming heterostructures with metals or semi-metals provides pathways for electrically contacting the vdW QWs, enabling active tuning of the resonance energies through carrier injection. The intersubband transitions in these vdW QWs have garnered growing interest and have been investigated using techniques such as infrared nano-imaging[18], photoluminescence excitation[24], magneto-transport[20] and resonant tunneling[25,26] techniques.

In spite of their unique spectral properties, the coupling of the intersubband transitions in vdW QWs with free-space photons has proven challenging, due to the mismatch between the out-of-plane polarization of the intersubband optical transitions and the in-plane polarization of the normal incident free-space photons[12,22]. To overcome the momentum mismatch, photonic cavities such as planar resonators[8] or metasurfaces[12] are often employed, but their use significantly increases device fabrication complexity. The bulky structure of these devices poses challenges for integration with other optical components on the chip. Furthermore, the weak oscillator strength between the lower subbands induced by typical electrostatic doping has rendered the observation of ISP propagation elusive[18,27]. Achieving high carrier doping within the quantum well, along with the use of out-of-plane optical probes, is essential for advancing the study of ISPs in vdW QWs.

Recently, tungsten oxyselenide, the native-dielectric created by oxidizing the top-most monolayer of WSe$_2$[28], has been shown to provide efficient charge transfer to graphene when a direct contact is formed[29]. This oxidation process is self-limiting, providing an efficient and controllable doping method that yields a carrier density nearly an order of magnitude higher than that achievable by electrostatic gating[28–30]. This approach has been demonstrated to efficiently modulate the carrier density in graphene and thereby allow engineering of plasmon polaritons[31]. Although promising for carrier density control, its application to the formation and study of ISPs remains unexplored. Herein, we report that by utilizing this native transition metal oxide (TMO) charge transfer strategy on few-layer WSe$_2$, we can substantially populate the higher energy subbands with carriers resulting in intersubband transitions with sufficient oscillator strength, suitable for reaching a polaritonic regime. With this system improvement, we experimentally demonstrate that the ISP propagation in WO$_x$/4L-WSe$_2$ heterostructure can be imaged optically, readily in real-space, with nano-scale resolution scattering-type scanning near-field optical microscopy (s-SNOM). We quantify the dispersion of the observed hyperbolic modes and demonstrate the electrical tunability of the ISPs by further applying gate bias to the heterostructure device. Finally, we in addition fabricate lithographically patterned oxide transfer doping layers to further provide an additional lateral quantum confinement to the ISPs. Within these nanoresonators, we demonstrate that devices exhibit hyperbolic rays traveling along conical trajectories and have a negative dispersion relation in the highly confined ISPs.

## Results
### Intersubband transition and native-dielectric-doping

The intersubband transition energies in the WSe$_2$ QWs can be approximated with an infinite square well potential with well width $L = Nd$, where $N$ is number of layers and $d$ is the monolayer thickness (Fig. 1a). The dispersion near the band edge at the $\Gamma$ point can be obtained from

$$E_\Gamma(k_z, \boldsymbol{k}) \approx -\frac{\hbar^2 k_z^2}{2m_{v,z}} - \frac{\hbar^2 k^2}{2m_{v,xy}}(1 + \zeta k_z^2) \tag{1}$$

where $m_{v,z}$ and $m_{v,xy}$ are the out-of-plane and in-plane effective masses, $\zeta$ is the nonlinearity parameter and $k_z \approx \frac{\pi n}{d(N+2v)}$[22]. The intersubband transition energies can be expressed as

$$|E_1 - E_2| = \frac{3\pi^2\hbar^2}{2m_{v,z}d^2(N+2v)^2} \tag{2}$$

and

$$|E_2 - E_3| = \frac{5\pi^2\hbar^2}{2m_{v,z}d^2(N+2v)^2} \tag{3}$$

We compare the intersubband transition energies calculated with this modified infinite square well model with density function theory (DFT) and find a good agreement with experimental values[21] (Fig. 1b). The intersubband transition energies in multilayer WSe$_2$ can be tuned from 636 meV to 39 meV by control of the number of layers. To engineer optical access to the intersubband states in the mid-infrared energy range, typical of a QCL source, we specifically choose a platform of a five-layer (5 L) 2H-WSe$_2$ homostructure to create a highly doped vdW multi-QW with accessible, partially occupied subbands. Figure 1c illustrates the fabrication process of WO$_x$-doped WSe$_2$ multilayers. In our method, the electronic band structure is, in essence, adjusted through the self-limiting oxidation of the top WSe$_2$ layer. We subject a dry-transferred WSe$_2$ flake to a UV-Ozone treatment, designed to convert the top layer into a TMO layer which has a high work function, while maintaining the atomic integrity of the underlying layers (see Methods and Supplementary Note 1). The significant work function difference between WO$_x$ and WSe$_2$ induces a surface charge transfer[32], promoting strong hole doping in the WSe$_2$ layers. The hole density in these $p$-doped WSe$_2$ multilayers is measured under ambient conditions using standard four-probe measurements (Supplementary Note 2). We find a hole density of $p = 0.93 \times 10^{13}$ cm$^{-2}$ at $V_{GS} = 0$ V, aligning with prior reports of doping in the $10^{13}$ cm$^{-2}$ range[29]. By applying a back gate voltage, $V_{GS}$, to the heavily doped Si substrate separated by the 300 nm silicon oxide layer, we can further increase this density to $p = 1.11 \times 10^{13}$ cm$^{-2}$ at $V_{GS} = -25$ V. The doping level achieved through the TMO layer process surpasses most electrostatic gating methods using solid dielectrics by an order of magnitude. Electrostatic approaches generally yield densities in the $10^{11}$ to $10^{12}$ cm$^{-2}$ range[33].

We calculated the expected electronic band structure modification from the top WO$_x$ layer onto the WSe$_2$ layers underneath, using density functional theory (DFT) incorporating spin-orbit coupling effects (Supplementary Note 2). Figure 1d, e presents the band structure of WO$_x$/4L-WSe$_2$ heterostructure calculated using this approach. These calculations elucidate the splitting of the highest valence band into multiple subbands at the $\Gamma$ point in the Brillouin zone, corresponding to the interlayer coupling. Upon $p$-doping, free holes partially occupy these subbands from the subband base to the Fermi energy ($E_F$), transitions can occur within the $|k| \le k_F$ range between a subband and a higher unoccupied subband. This is manifested as an absorption peak. To investigate the influence from the WO$_x$ layer, selenium atoms in

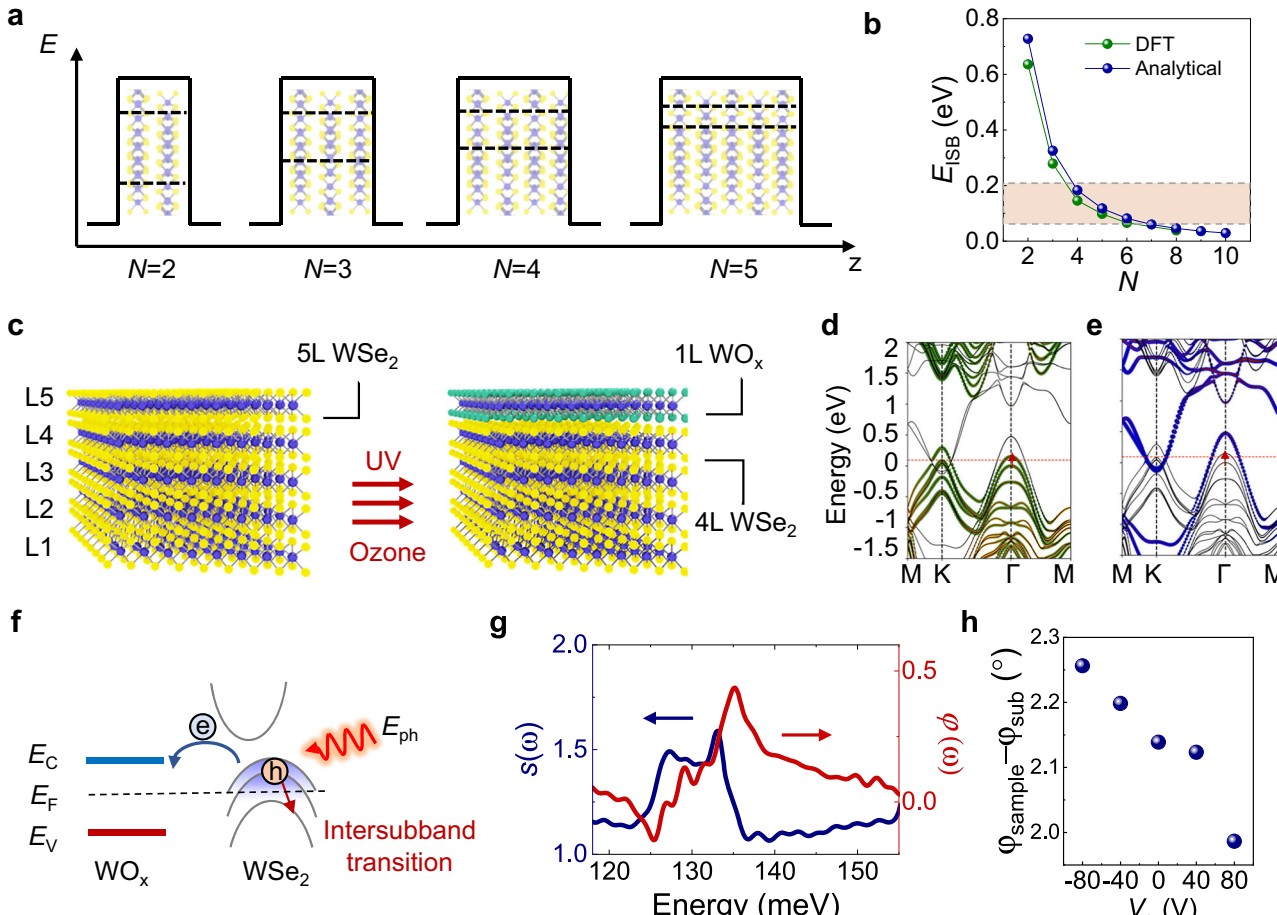

**Fig. 1 | Intersubband transition in WSe₂ and WOₓ heterostructure. a** Schematic illustration of subbands in the van der Waals (vdW) material quantum wells with different thickness defined by the number of layers $N$. Black dashed lines illustrate the subband energy positions. **b** Theoretical calculation of the intersubband transition energy of the first and second subband as a function of $N$ for holes in the valence band using density function theory (DFT) (green circles) and modified infinite quantum well model, respectively. Red shaded area indicates the energy coverage of our mid-infrared laser. **c** Schematics of the doping process. Ultraviolet-ozone oxidation transforms the topmost layer of the 5L-WSe₂ into transition metal oxides (TMOs) resulting in WOₓ/4L-WSe₂. The oxidation process effectively doped the underneath WSe₂ layers. DFT calculated band structure for WOₓ/4L-WSe₂

heterostructure showing contribution from 4L-WSe₂ in green (**d**) and top WOₓ layer in blue (**e**). Red dashed lines indicate the Fermi energy $E_F$. Red solid arrows indicated the optical transition. **f** Illustration of charge transfer between WOₓ and WSe₂ as the result of the work-function mismatch. $E_C$ is conduction band edge and $E_V$ is valence band edge; Excitation light with out-of-plane polarization can excite the charge carriers from the lower state to the upper excited state if excitation photon energy $E_{ph}$ is in resonant with the $E_{23}$ intersubband transition energy. **g** Nano Fourier transform infrared spectrum of the WOₓ/4L-WSe₂ near-field amplitude $s(\omega)$ and phase $\varphi(\omega)$. **h** Phase contrast between the sample and substrate as a function of gate voltage $V_g$.

the topmost layer were substituted with oxygen, and the ionic positions subsequently re-optimized. As depicted in Fig. 1d, e, this modification of the electronic states results in an altered band structure at the $\Gamma$ point, displaying distinct subband splitting at mid-IR energies. Our calculation suggests that when the top layer (L5) is oxidized, the symmetry is broken and the top valence subband comes almost exclusively from the L5. The 2nd valence subband then comes mostly from L4 and L3 while the 3rd valence subband from L2. In order to maximize the coupling of light to intersubband transitions, it is desirable to populate holes in the second subbands in a few-layer WSe₂, which requires carrier density more than ~$10^{13}$ cm⁻² and has been challenging to reach by electrostatic gating alone. In contrast, using the TMO layer we can easily generate the sufficient carrier density with $E_F$ positioned within the valence band. As illustrated in Fig. 1f, when the heterostructure is excited by photon with the out-of-plane polarization whose energy is resonant with the transition energy between the subbands, an intersubband transition between the second and third subbands can be observed.

## Near-field imaging of the intersubband polaritons

An s-SNOM equipped with a broadband pulsed laser is utilized to carry out the nano-Fourier-transform infrared spectroscopy (nano-FTIR). The infrared excitation laser beam is focused on the metallized tip of the atomic force microscope (AFM), generating an enhanced optical field that interacts with the heterostructure sample beneath. With that configuration, we are able to characterize the intersubband transitions in WOₓ/4L-WSe₂ at near-field with out-of-plane excitation polarization. The scattered light from the tip is collected, and the near-field amplitude and phase are recorded from the sample flake. This data is calibrated against a gold substrate to eliminate any instrumental responses. The near-field results provide insights into the sample's complex permittivity, optical conductivity, and its absorption characteristics under the excitation energy $E_{ph}$ (Fig. 1g). Notably, while the dielectric response of WSe₂ remains featureless from 120 meV to 160 meV[34], WOₓ/4L-WSe₂ heterostructure exhibits a distinct absorption peak, clearly indicative of the predicted intersubband transition. To avoid interference from SiO₂ phonons, typically prominent around 140 meV, an Au/Si substrate without an SiO₂

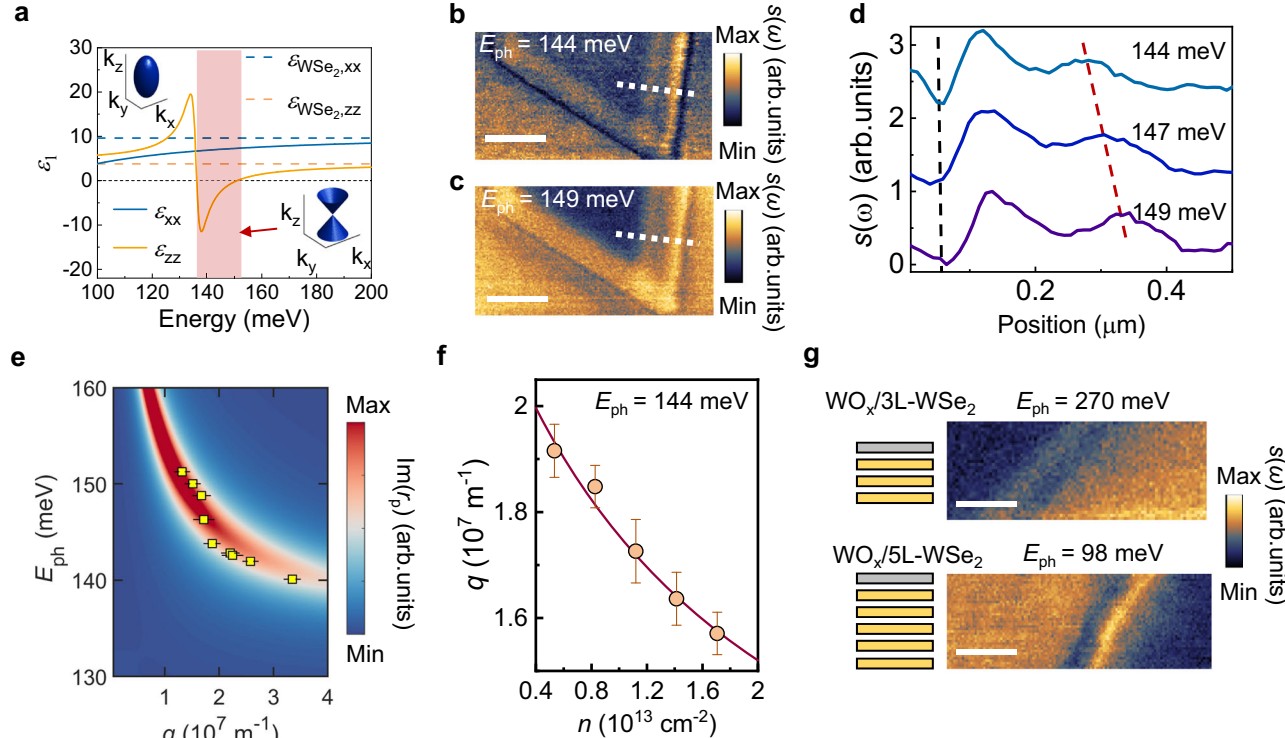

**Fig. 2 | Nano-imaging of intersubband transition and hyperbolic plasmon polariton in WO$_x$/4L-WSe$_2$. a** Permittivity of the WSe$_2$ (dashed lines) and with high carrier density under charge transfer (solid lines). Inset: the isofrequency surface modified from ellipse to a hyperboloid in the red-shaded region. Images of near-field scattering amplitude $s(\omega)$ for WO$_x$/4L-WSe$_2$ with excitation energy $E_{ph} = 144$ meV (**b**) and $E_{ph} = 149$ meV (**c**). The layer thickness is $d \sim 4.5$ nm. Scale bar: 500 nm. **d** Line traces of the interference pattern from the edge-reflected plasmon polariton taken from the near-field scattering amplitude image as illustrated in (**b**) and (**c**) (white dashed lines). Black dashed line marks the edge of the sample and red

dashed line marks the peak position. **e** Dispersion of the intersubband polariton (ISP) in WO$_x$/4L-WSe$_2$. Color plot is calculated using the Fresnel reflection coefficients Im($r_p$). Yellow squares are experimental data. The error bars are determined from the fitting uncertainties of the fringe periodicities. **f** The extracted real part of momentum $q$ as a function of carrier density $n$ for WO$_x$/4L-WSe$_2$. The red solid line is the fit to the data using the relation $q \approx 1/n$. The error bars are determined from the fitting uncertainties of the fringe periodicities. **g** Images of near-field scattering amplitude $s(\omega)$ for WO$_x$/3L-WSe$_2$ and WO$_x$/5L-WSe$_2$ with excitation energies $E_{ph} = 270$ meV and $E_{ph} = 98$ meV, respectively. Scale bar: 250 nm.

layer was utilized. We further explore the nano-FTIR result by fitting the data with modulated scattered field where tip-sample interaction and complex permittivity described as Lorentz model are taken into consideration (Supplementary Note 4). We then demonstrate inter-subband absorption control by analyzing the phase contrast between the sample and the substrate $\varphi_{sample}$-$\varphi_{sub}$ while changing the carrier density using a gate bias. The interferometric detection of the near-field optical response can decouple the amplitude and phase components from the scattered field as $E_{scat} \propto S e^{i\varphi} E_{in}$, where $S$ is the relative amplitude and $\varphi$ is the phase shift[35]. Hence, the relative phase shift is a measurement of absorption at the near-field. As shown in Fig. 1h, we can clearly see that the absorption decreases when we inject electron into the sample to decrease the hole carrier density. (Note the electrostatic bias cannot tune the carrier density to the charge neutrality point due to the high doping level from the charge transfer of the WO$_x$).

According to the nano-FTIR spectrum, the $\varepsilon_{imag}$ should exhibit strong resonant-like features along the out-of-plane crystal direction, implying the anisotropic dielectric permittivity should satisfy $\varepsilon_{real,i} \times \varepsilon_{real,j\neq i} < 0$, since the real and imaginary parts of the dielectric function are connected through the Kramers–Kroing relation[36]. Due to the different value of the effective masses and dielectric screening in the in-plane and out-of-plane directions, the highly doped WO$_x$/4L-WSe$_2$ heterostructure exhibits strong anisotropy in the frequency range (Fig. 2a). The heterostructure behaves like a dielectric in-plane, where $\varepsilon^{x,y} > 0$. For the out-of-plane direction, the heterostructure behaves like metal under the high carrier density conditions, showing a strong intersubband absorption with $\varepsilon^z < 0$ at the mid-IR frequency.

We note that this phenomenon is fundamentally different from recently demonstrated bound carrier excitation in bulk WSe$_2$ near the interband optical transition energy[37]. In the bulk WSe$_2$ crystal, the subbands remain degenerate, and hyperbolic behavior emerges only under non-equilibrium conditions, in contrast to the ISPs studied here.

To effectively couple intersubband transitions to free-space photons and excite ISPs in the WSe$_2$ quantum wells, we employed s-SNOM with both a continuous-wave QCL and a pulsed optical para-metric oscillator laser to sweep across the transition energies. This configuration introduces an out-of-plane light component that drives the intersubband transitions within the heterostructure, thereby giving rise to ISPs. In order to reveal the ISP propagation in the hetero-structure, we now investigate the polariton dispersive dynamics interferometrically using the real-space near-field nano-imaging tech-nique. Figure 2b, c presents data collected at different excitation energies, $E_{ph} = 144$ meV and $E_{ph} = 149$ meV, respectively. Oscillations of the near-field amplitude $s(\omega)$ are observable at the flake edge. These fringes arise from the interference between the tip-launched polariton and the propagated polariton reflected at the sample edge. By exam-ining the cross-sectional profile perpendicular to the sample edge, we can extract the periodicity of the bright-dark fringes, which corre-sponds to half of the polariton wavelength ($\lambda_{ISP}/2$). As the excitation energy increases from 144 meV to 149 meV, the polariton wavelength increases from $317 \pm 10$ nm to $453 \pm 12$ nm (Fig. 2d). The propagation length of the ISP is fitted to be 0.8 μm (Supplementary Note 5). The lifetime of the ISP is calculated as $\tau_{ISP} = \frac{L}{v_g} = 0.4 ps$ (Supplementary Note 10). Using the same method, we systematically extract the polariton wavelengths at various tuned excitation energies from the

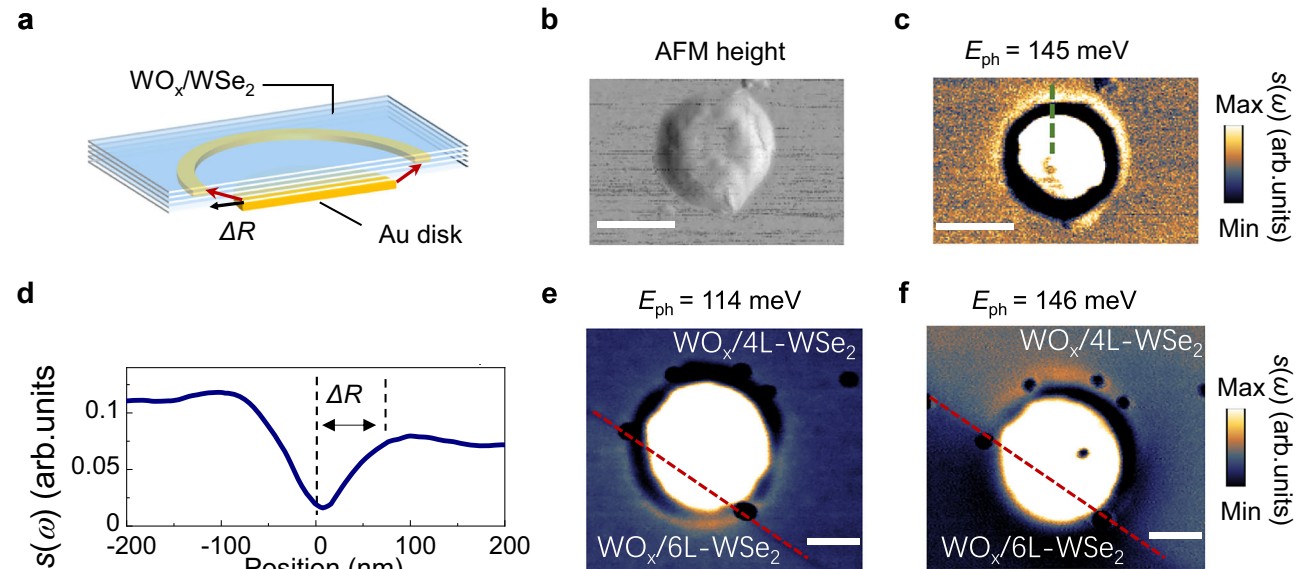

**Fig. 3 | Hyperbolic ISP revealed by nano-imaging on Au nanodisk. a** Schematic of hyperbolic polaritons launched at the edge of the Au nanodisk, which travel along conical trajectories and form a bright ring. The distance between the ring and the edge of the Au nanodisk is $\Delta R$. Topography (**b**) and near-field scattering amplitude $s(\omega)$ image of $WO_x/4L$-$WSe_2$ plasmonic cavity with diameter $D = 300$ nm and excitation energy $E_{ph} = 145$ meV (**c**). Scale bar: 300 nm. **d** Line profile of the measured near-field amplitude taken at the dashed line in (**c**). Near-field scattering amplitude $s(\omega)$ image of sample on the Au nanodisk that is partially covered by $WO_x/4L$-$WSe_2$ and $WO_x/6L$-$WSe_2$ simultaneously and excited with $E_{ph} = 114$ meV (**e**) and $E_{ph} = 146$ meV (**f**), respectively. Red dashed lines indicate the boundary. Scale bar: 150 nm.

nano-imaging data. The confinement factor $\lambda_0/\lambda_{ISP}$ reaches up to 27, comparable to that of plasmon polaritons in graphene and phonon polaritons in h-BN, while still smaller than the acoustic plasmon in graphene[38]. We visualize the dispersion relations with a false color mapping of the imaginary part of the reflection coefficients Im ($r_p$), representing the photonic density of states as a function of wave vector $q = 2\pi/\lambda_{ISP}$ (Fig. 2e and Supplementary Note 6). This inverse relationship between energy and momentum confirms the Type I hyperbolic polariton dispersion, where $\varepsilon^z < 0$ and $\varepsilon^{x,y} > 0$.

As noted, in addition to the charge transfer process via TMO layer, the carrier density can be further tuned. We demonstrate the electrical control of the polaritons by systematically tuning of the intersubband transitions in the valence band via back-gating. Since the charge transfer method between the TMO layer and $WSe_2$ drives a much higher carrier density than that possible via tuning by electrostatic gating, we can vary the carrier density modestly in the $p$-type doping regime. Figure 2f shows the ISP momentum $q$ as a function of the back-gate voltage $Vg$ at a representative energy $E_{ph} = 144$ meV. The magnitude of $q$ is approximately proportional to $n^{-1}$, where higher hole density leads to increased negative permittivity[39]. The evolution of $q$ with $Vg$ further confirms the hole doping in the $WO_x/4L$-$WSe_2$ structure and the observed negative dispersion relation (Supplementary Note 8). Such active tunability is generally absent in the natural hyperbolic materials such as h-BN[40] and $MoO_3$[41]. The intersubband transition energy can be controlled with the thickness of the vdW layers. We also demonstrated the nano-imaging of the ISP in both $WO_x/3L$-$WSe_2$ and $WO_x/5L$-$WSe_2$ heterostructure at excitation wavelength $E_{ph} = 270$ meV and $E_{ph} = 98$ meV, respectively (Fig. 2g). This finding aligns well with the theoretical calculation shown in Fig. 1b.

### Measurement of the hyperbolic propagation
We explore the presence of hyperbolic ISP by positioning the $WO_x/4L$-$WSe_2$ heterostructure on the 50 nm Au disk (Fig. 3b). The conical-shaped rays launched by the edge of the Au disk will reach the top surface of the heterostructure forming bright rings separated by the edge (Fig. 3a). The distance between the bright ring and the

electrostatic edge, denoted as $\Delta R$, is modulated by the isotropic components of the dielectric tensor, with the relationship $\Delta R/d = |\tan\theta| = i\sqrt{\varepsilon_{xy}}/\sqrt{\varepsilon_z}$, where $d$ represents the heterostructure's thickness, and $\theta$ is the angle from the surface normal[42,43]. Directional propagation of the hyperbolic ISP along the resonance cone is observed at $E_{ph} = 145$ meV (Fig. 3c). The line-profiles taken from the s-SNOM amplitude image reveal that the radial distance $\Delta R$ is significantly smaller than that observed in bulk hyperbolic materials, due to the thickness of the heterostructure as thin as $d \sim 4.5$ nm (Fig. 3d). We further verify the hot-ring feature for different layer number samples by transferring the heterostructure with different thickness onto the same Au disk. When the excitation energy matches the $E_{23}$ transition energy of the $WO_x/6L$-$WSe_2$ at $E_{ph} = 114$ meV, we can see only the bottom half lights up (Fig. 3e). When we increase the excitation energy to $E_{ph} = 146$ meV, the upper half lights up as the intersubband transition energy increases with the fewer layer numbers (Fig. 3f). The layer-dependent hyperbolicity highlights ISPs as a promising addition to the family of hyperbolic polaritons, complementing other recently demonstrated phenomena such as topological transitions[44], negative refraction[45,46], and low-loss propagation[47].

### Intersubband polaritons in nanoresonators
We further manipulate the hyperbolic ISP by fabricate the $WO_x/4L$-$WSe_2$ heterostructure into nanoresonators where the polariton modes are confined by the circular boundaries. Figure 4a illustrates the schematic of an s-SNOM measurement conducted on focus ion beam (FIB) patterned $WO_x/4L$-$WSe_2$ nanoresonators with an underlying gold mirror. By eliminating the need for a polymer mask in the etching process, direct FIB fabrication keeps the heterostructure surface uncontaminated, minimizing the risk of residue deposition or chemical interaction commonly associated with mask-based techniques, thereby preserving the material's intrinsic properties and optimizing the quality of subsequent device performance. Additionally, a gold mirror is deposited on the substrate to minimize radiative decay into the silicon substrate, thereby resulting in lower loss and longer lifetime (Supplementary Notes 9 and 10). To explore

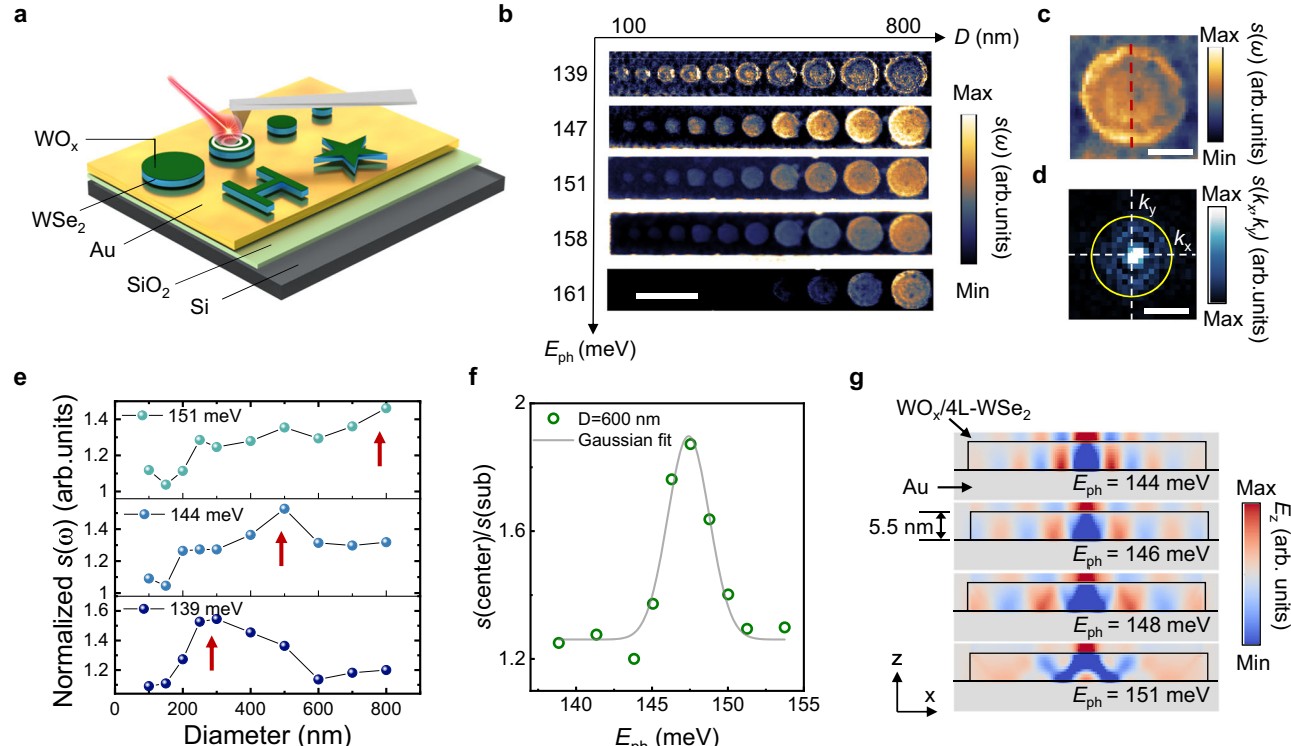

**Fig. 4 | Intersubband polariton confined in WO$_x$/4L-WSe$_2$/Au nanoresonators.** **a** Schematics of WO$_x$/4L-WSe$_2$/Au nanoresonators fabricated by focused ion beam on gold substrate. **b** Near-field scattering amplitude $s(\omega)$ images of an array of WO$_x$/4L-WSe$_2$/Au nanoresonators with different diameters (100 nm to 800 nm) and different shapes. The images are measured at a series of excitation energies from 139 meV to 161 meV. Scale bar: 1 µm. **c** Near-field scattering amplitude $s(\omega)$ image of WO$_x$/4L-WSe$_2$/Au nanoresonator with diameter $D = 600$ nm. The excitation energy is $E_{ph} = 146$ meV. Scale bar: 250 nm. **d** The corresponding Fourier transform image of (**c**). $k_x$ and $k_y$ are wavevectors in $x$ and $y$ direction. Scale bar: 40 $k_0$, where $k_0$ is the momentum of light in free space. **e** Normalized near-field scattering amplitude as a function of diameters of the resonators with excitation energy at $E_{ph} = 151$ meV (top panel), $E_{ph} = 144$ meV (middle panel) and $E_{ph} = 139$ meV (bottom panel). Red arrows help to identify the peak value of $s(\omega)$. **f** Normalized amplitude as a function of excitation energy for the WO$_x$/4L-WSe$_2$/Au nanoresonator with diameter $D = 600$ nm (green circles). Gray solid line is the Gaussian fitting. **g** Finite-difference time-domain simulation of the out-of-plane field $E_z$ in WO$_x$/4L-WSe$_2$/Au nanoresonator with different excitation energies of $E_{ph} = 144$ meV, $E_{ph} = 146$ meV, $E_{ph} = 148$ meV and $E_{ph} = 151$ meV, respectively. The diameter of the nanoresonator is $D = 600$ nm.

the relation between the resonant energy and the diameters of the nanoresonators, we patterned disk nanoresonators with diameters ranging from 100 nm to 800 nm. The propagation of the ISP in the nanoresonators can be described by the wave equation $\rho(i\omega)^2 = \nabla^2 \rho v_p^2(\omega)$, where $\rho$ denotes the integrated two-dimensional charge density and $v_p$ is the group velocity[48]. For a disk-shaped resonator, the solution of $\rho$ can be expressed in terms of Bessel functions, with the eigenvalues $k_{sn}$ determined by the disk diameter. Consequently, because the ISP can resonate only with resonators of specific sizes at a given excitation photon energy, tuning the $E_{ph}$ enables the systematic mapping of the relationship between disk diameter and the ISP resonant modes (Fig. 4b).

To quantitatively study the confined mode in the resonators, we analyzed each disk resonator at different excitation energies. The near-field amplitude image of a representative nanoresonator with a diameter of $D = 500$ nm is shown in Fig. 4c, where a standing-wave pattern with multiple fringes emerges[49,50]. Figure 4d presents the Fourier transform $|s(k_x, k_y)|$ of the near-field amplitude image from Fig. 4c, revealing the iso-frequency contours for the ISP at 146 meV. The in-plane permittivity of the WO$_x$/4L-WSe$_2$ heterostructure is isotropic. We then extract the near-field amplitude response from the standing wave in the resonator at different excitation energies (Fig. 4e). The signal was normalized to the background at the surrounding gold surface to compensate for variations in the laser power and optical alignment in each measurement. At $E_{ph} = 139$ meV, the near-field amplitude

shows a peak $D = 300$ nm. The size of the resonator imposes a constraint on the ISP wavelength confined in the resonator. With increasing $E_{ph}$, disk diameter exhibiting a stronger scattered near-field response also increases due to the longer polariton wavelength, aligning with the hyperbolic dispersion relation observed in the unpatterned heterostructures. We then investigate the field concentration and the resonance behavior of the confined ISP by analyzing a resonator with $D = 600$ nm under various excitation energies. Figure 4f shows the near-field optical response as a function of the excitation energy. The signal is normalized to the substrate background to eliminate the influence from the power stability and detection efficiency. We fit the peak with Gaussian function with center energy at 147 meV and FWHM = 3.2 ± 0.3 meV. The quality factor is then calculated as $Q = \omega/\text{FWHM} = 45.9$, where $\omega$ is the resonant frequency[48,51].

We finally conducted a numerical simulation for the resonator with same diameter of $D = 600$ nm to further explore the polaritons propagation in WO$_x$/4L-WSe$_2$ nanoresonators. The $z$-component electric field distribution, mapped at various excitation energies in Fig. 4f, shows that as the $E_{ph}$ increases from 144 meV to 148 meV, the distance between the bright maxima widens, following the negative dispersion relation. At an $E_{ph} = 151$ meV, the polariton wavelength $\lambda_{ISP} = 478$ nm exceeds the nanoresonator's radius, precluding effective ISP mode confinement. Our FDTD simulations show that the highly confined modes have an effective mode volume $V$ of $2 \times 10^{-6}/\lambda_0^3$, respective to the free space photon wavelength $\lambda_0$, which quantifies *the strong*

*spatial confinement* of light in the disk nanoresonators[52]. These observations highlight that the structural refinement of nanoresonators via FIB patterning facilitates enhanced control over polariton modes, resulting in a higher quality factor and improved mode confinement. Importantly, there remains potential to further optimize the cavity design to achieve even greater polariton confinement with a significantly reduced mode volume[53].

## Discussion

We report the creation and experimental observation of ISPs in high-density *p*-type doped $WO_x$/$WSe_2$ QW heterostructures. These widely tunable ISPs are systematically studied using spatially and spectrally resolved near-field microscopy. Unlike phonon-polaritons in h-BN[40] or molybdenum oxide ($MoO_3$)[41], where the polariton properties are fixed by the atomic arrangement of the crystal structure, the ISPs in atomically thin vdW QW heterostructures can be actively tuned by varying the carrier density with electrostatic doping in addition to the passive band structure engineering via doping using self-limited oxidation layers. Intersubband transitions are observed in other 2D semiconductor materials such as $MoS_2$[18,19] and the monolayer TMO strategy is transferable to other materials with lower work function to provide a path to high-density *p*-type doping[31]. We note that the ISPs excited can exist in much broader energy ranges than other types of polaritons, potentially from near IR to THz, due to the various subband spacing energies possible (Fig. 2g and Supplementary Note 7). Furthermore, with high-quality wafer-scale TMD crystals becoming available, this 2D vdW QW platform potentially allows for a wide variety of applications featuring tunable hyperbolic polaritonic devices.

## Methods

### Sample fabrication

We exfoliated thin $WSe_2$ flakes onto $SiO_2$ substrates using Scotch tape from a bulk commercial source (HQ Graphene). Suitable five-layer flakes were identified via optical contrast, using a "staircase" flake on the same chip to reliably distinguish between crystals of various thicknesses. We subsequently dry-transferred the flakes onto a separate $SiO_2$ substrate coated in Ti/Au (5 nm/50 nm). Devices with contacts were transferred onto a bare $SiO_2$ substrate with a heavily doped Si back gate for density control. The contacts consisted of 20 nm/70 nm of Pd/Au deposited directly on top of the $WSe_2$ flake patterned via electron beam lithography with a polymer mask, following the recipe in ref. 1. For all devices, we oxidized the top TMD surface as the very last step just before measurements to ensure a high-quality oxide and clean off any residues. Oxidation was performed in a Samco UV & Ozone Stripper with a plate temperature of 100 °C for 1 h. All measurements were carried out at room temperature.

### Carrier density measurement

We fabricated a separate device with various electrodes to electrically characterize the induced properties from the top oxide layer. We applied a constant voltage bias of 0.25 V and swept the Si gate $V_{Si}$ with Keithley 2400 sourcemeters. We measured the 4-terminal (4t) voltage $V_{4t}$ between source and drain with an Agilent 34401A 6½ Digit Multimeter. We extract the field-effect mobility $\mu_{FE}$ of the $WO_x$/4L-$WSe_2$ to be $0.9 \pm 0.1$ cm²/Vs by using $\mu_{FE} = \frac{L_{4t}}{CWV_{4t}} \frac{dI}{dV_{Si}}$, where $L_{4t}$ is the separation between the 4t voltage probes, $C$ is the 285 nm $SiO_2$ dielectric gate capacitance, $W$ is the estimated channel width, and $I$ is the source-drain current. We calculated $\frac{dI}{dV_{Si}}$ from the linear regime of the field effect transistor. To calculate the hole density, we use the relation $p(V_{Si}) = \frac{IC}{e} \left(\frac{dI}{dV_{Si}}\right)^{-1}$, where $e$ is the elementary charge.

### Near-field optical measurement

The nano-imaging experiments were performed using commercial s-SNOM (Attocube systems AG). The system is equipped with QCL laser and OPO for the nano-imaging within mid-IR range and broadband DFG laser for nano-FTIR measurement. Platinum silicon coated AFM tips were used in the s-SNOM with a typical radius of 20 nm. The s-SNOM was operated under the tapping mode with a tapping frequency around 240 kHz. We use pseudo-heterodyne interferometric detection module to extract the near-field amplitude and phase signals. The background signal is suppressed by demodulation of the near-field signal at the third harmonics of the tapping frequency. All the near-field optical measurements are done at room-temperature.

### Numerical simulations

The numerical simulations of the propagation of ISPs and nanoresonators were carried out by the FDTD simulation software (FDTD solutions). The dipole sources are used to excite the polariton and perfect metal layer boundary condition is used to simulate the near-field modes.

## Data availability

The Source Data underlying the figures of this study are available at https://doi.org/10.5281/zenodo.17288925. All raw data generated during the current study are available from the corresponding authors upon request.

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

## Acknowledgements

This work was partially performed at CNS support by the National Science Foundation (NSF) under award ECCS-2025158. This work was partially carried out at the USTC Center for Micro and Nanoscale Research and Fabrication. Y.L. was supported by the Southeast University Interdisciplinary Research Program for Young Scholars (Fundamental Research Funds for the Central Universities, 2242025F10008). Y.L acknowledges the support from National Natural Science Foundation of China (Grant No.9247710104). Y.L and J.W. acknowledge the support from Center for Fundamental and Interdisciplinary Sciences. D.D. acknowledges the support from Samsung Electronics. A.M.V. is supported by AFOSR (FA2386-21-1-4086). P.K. and E.K. acknowledges the support from ARO MURI (W911NF-21-2-0147). D.T.L is supported by NSF Award No. DMR-1922172. E.K. acknowledges the support from ARO (W911NF-21-1-0184). Calculations were performed on the FASRC Cannon cluster supported by the FAS Division of Science Research Computing Group at Harvard University. Authors thank Wei Li and Weikang Wang for the help on nano-FTIR measurements as well as Gabriel Schleder and Daniel Bennett for helpful discussions.

## Author contributions

Y.L., H.P., P.K., and W.L.W. conceived the experiment. S.L. carried out the crystal growth. D.D., A.M.M.V., S.L. fabricated the device samples. H.K.N. and J.W. fabricated the Au disk samples. K.W. and T.T. provided the h-BN crystal. D.T.L. and E.K. performed the DFT calculation. Y.L. carried out the s-SNOM experiments and analyzed the data with P.K. and W.L.W. and Y.L. performed theoretical calculations and FDTD simulations for the polariton. All authors discussed results and wrote the manuscript.

## Competing interests

The authors declare no competing interests.
