## [Transparent Peer Review file · Nature Communications]

Observation of hyperbolic intersubband polaritons in native-dielectric-doped van der Waals semiconductor quantum wells

Corresponding Author: Professor Yue Luo

Version 0:

Reviewer comments:

Reviewer #1

(Remarks to the Author)

Y. Luo and co-authors report the near-field study of polaritons in highly doped multilayer WSe₂ which is achieved by modifying the top layer into a monolayer of tungsten oxide. Authors also demonstrate the hyperbolic nature of these polaritons by near-field probing of patterned disc-shaped nanoresonators made by two different fabrication methods. Unfortunately, I cannot recommend this manuscript for publication in Nature Communications due to the insufficient novelty, numerous unsubstantiated claims, and instances of incorrect scientific analysis. Below I list the detailed description of the most significant issues of this manuscript:

1. Mid-infrared hyperbolic polaritons in doped WSe₂ have been demonstrated in [Science 371, 617-620 (2021)], which is not even cited by the authors. Without detailed comparison, I do not see how the demonstrated polaritons are different from those demonstrated in the Science paper.
2. The mechanism of the intersubband polaritonic excitation is not explained, thus it is not clear how the experimental results demonstrate the intersubband nature of polaritons. For example, why it is important to populate the second subband as stated in line 137? And why the incident light polarization is relevant for the transitions between the subbands as stated in line 140-141?
3. The nano-FTIR absorption peak by itself does not indicate the presence of the specific intersubband transitions (at what energy?) as stated in line 150 and 154. For example, the near-field absorption peak can be associated with other intraband transitions with the same energy, varying ohmic loss in doped WSe₂ due to charge screening, changes in the near-field reflection coefficient of the top WO layer caused by varying carrier concentration, or even unknown phonons in the newly studied heterostructure.
4. The near-field fringes analysis presented in Supplementary Note 5 seems to be incorrect. First of all, the near-field inset in S.Fig. 3 does not correspond to the fitted line profile. Furthermore, the near-field images in the main text Fig. 2 only show a single fringe, while S.Fig. 3 shows at least two fringes. Where did this data come from? But most importantly, the weak near-field signal with only 1-2 fringes cannot be correctly fitted by damped sine or cosine functions because the zero coordinate is unknown. This ambiguity introduces a very big error in the fitted parameters. The proper way to fit weak near-field fringes is to use complex near-field data as was demonstrated in [Nat. Commun. 13, 1374 (2022)]. Because of this fitting error, authors greatly overestimated the polariton quality factor, claiming $Q = 20$ which is typical for phonon-polaritons in hBN [Nature Mater. 17, 134–139 (2018)] and four times exceeds $Q \sim 5$ of graphene plasmons [Nature 487, 82–85 (2012)]. Although the fitting of polariton wavelength is less sensitive to the zero-coordinate problem, I encourage authors to verify all dispersion data extracted from near-field images by using proper fitting method.
5. There can be no radiative loss for such a high-momentum polaritons even on Si substrate as stated in line 233.
6. The quality factor definition in line 259 is invalid for near-field amplitude because it has strongly non-linear dependency

on the excitation energy which is not compensated by normalization.

Minor concerns:

- Quality of figures must be improved. Some panels are too small to see the claimed features (for example, where are the green/yellow and blue/red points in Fig. 1d and e as stated in line 131-132?).
- Contents delivery and manuscript structure must be improved. The theoretical introduction in lines 97-104 lacks definitions and feels too technical for the main text. Discussion of SNOM operation principle is missing, which is crucially important to understand nano-FTIR data and near-field images.
- Text contains many typos.

Reviewer #2

(Remarks to the Author)

Luo et al. present a compelling and timely study on the realization and manipulation of intersubband polaritons in van der Waals quantum wells. The innovation of this work lies in the use of a self-limiting native oxide to achieve high-density hole doping in few-layer WSe₂, which represents a significant materials engineering achievement. This technique successfully enables strong intersubband transitions, providing a robust platform for investigating mid-infrared polaritonic phenomena. The subsequent demonstration of propagating hyperbolic ISPs, their electrical tunability, and their confinement in nanoresonators constitutes a substantial advance for the field of nanophotonics. The work is well-structured, the data is generally of high quality, and the findings hold promise for the development of tunable mid-infrared devices.

While the research presents an important and timely advancement, several key aspects of the analysis and interpretation require further clarification and strengthening to fully substantiate the central claims of the manuscript. I recommend publication after the authors have addressed the following major points.

1. The observation of hyperbolic ISPs in this work is an important discovery. However, the reviewers have some questions regarding this definition. Typically, polaritons in semiconductors are exciton-polaritons, whereas the authors classify this mode as an intersubband plasmon-polariton. In addition, how do the authors distinguish this polariton mode from a waveguide mode? The reviewers hope the authors can provide further analysis and clarification regarding these questions.
2. The origin of the transition is central to the paper, but its depiction in Figure 1 could be improved. As presented, it is difficult to visually identify the specific subbands involved in the measured mid-IR transition from the DFT calculations (Fig. 1d-e). We strongly suggest adding graphical indicators (e.g., arrows) directly onto the band structure in Figure 1d to illustrate the transition between the second and third valence subbands at the Γ point.
3. The claim that the 'hot-ring' feature in nanodisks (Fig. 3) is conclusive proof of conical hyperbolic ray propagation requires more rigorous evidence, as similar ring-like features can be produced even in isotropic resonators (Adv. Funct. Mater. 2019, 29, 1904662).
4. The reliance on permittivity values from s-SNOM fitting warrants a more critical discussion, as extracted values can diverge from the actual dielectric function (Nature Reviews Materials 2025, 26). We suggest the authors justify their fitting procedure and validate the results with other measurements, such as ellipsometry or far-field optical characterization.
5. The real-space images in Figures 2 and 3 are primary evidence for polariton propagation, but the interference fringes have limited visibility. This may introduce uncertainty in the extracted wavelength. To strengthen the evidence, could the authors provide additional images with improved signal-to-noise ratio?
6. The reviewers find the variation in signal intensity with disk size, as observed in Figure 4b, to be a very interesting phenomenon. However, the current explanation for this observation is too brief. The reviewers request that the authors provide a more detailed and thorough explanation.
7. The manuscript would benefit from a more thorough discussion of its relationship to existing literature. Specifically, the use of WO_x for charge-transfer doping to tune polaritons has been demonstrated in graphene plasmonic systems (Nature Materials 22, 838–843, 2023). The authors should explicitly discuss this work and further clarify the novelty of their WSe₂-based intersubband polariton platform. Furthermore, to properly contextualize their findings, the authors should incorporate citations to other recent key developments in hyperbolic polaritons, such as studies on topological transitions in tunable materials like graphene-based heterostructure, demonstrations of negative refraction, and the discovery of hyperbolic plasmons in the visible range in novel bad metals.

Minor Errors:

1. Regarding the use of italics on lines 113 and 124, the intended meaning or emphasis is unclear to the reviewers. Please clarify the purpose of using italics in these instances.
2. The meaning of the dashed line in Figure 1a is not explained in the caption. Please add a definition.
3. There is an inconsistency in the formatting of the variable for the number of layers, N. For example, it appears non-italicized on line 314 but italicized on line 312. Please ensure uniform formatting throughout the manuscript.
4. The method used to determine the error bars in all figures should be explicitly stated.
5. The black dashed line is missing in Figure 2d.

Reviewer #3

(Remarks to the Author)
report attached

Reviewer #4

(Remarks to the Author)

Reviewer #5

(Remarks to the Author)

Version 1:

Reviewer comments:

Reviewer #1

(Remarks to the Author)

Report is attached.

Reviewer #2

(Remarks to the Author)

My co-reviewer and I have carefully reviewed the revised manuscript and the authors' responses to the previous comments. The authors have appropriately addressed the concerns raised in the first round, and the manuscript has been significantly improved. We believe this work is highly valuable and timely. The overall quality of the work is high, and the results are clearly presented.

I have only a few minor suggestions for further clarification:

1. In the captions of Figures 2e, 2f, and 3f, it would be helpful to include an explanation of the origin and calculation method of the error bars.
2. In Figure 3f, the alignment of the panel label "f" could be adjusted to match the order and style of the other figure labels.
3. In Figures 3b and 3c, the boundaries of the structures are currently depicted using a combination of white solid lines and colored dashed lines, which may slightly interfere with the presentation of the data.
4. In Figures 3b–3e, the colors of the marker lines and the data lines do not correspond optimally. For my understanding, the purple dashed line in Figure 3c might be better represented as a dark blue dashed line to improve consistency with the other panels.

These are minor points and do not affect the overall validity or significance of the work.

In conclusion, We are satisfied with the revisions and recommend acceptance of the manuscript for Nature Communications.

Reviewer #3

(Remarks to the Author)

The authors have adequately addressed all of my previous comments. In particular, they have corrected and improved Figure 3 by performing a new experiment that clearly demonstrates out-of-plane ray propagation of ISP polaritons. In light of the revised manuscript, I could consider its publication in Nature Communications after the authors have implemented two minor improvements listed below:

1. The authors demonstrate that the gold substrate enhances ISP propagation, leading to longer polariton propagation lengths and lifetimes. Such extended lifetimes (2.6 ps), comparable to those reported for hBN or MoO₃, represent a significant result for practical applications, such as the structure presented in Figure 4. I encourage the authors to emphasize this point more explicitly, for instance by reporting the lifetime value of this structure either in the main text or in Supplementary Note 9.
2. In Fig.3 the expression $\Delta R/d = |\tan\theta| = i \sqrt{\epsilon_{xy}\epsilon_{yz}}$ is employed in the analysis. The authors should compare the experimentally extracted value with the one obtained from substituting the permittivity values.

Reviewer #4

(Remarks to the Author)

Reviewer #5

(Remarks to the Author)

Version 2:

Reviewer comments:

Reviewer #1

(Remarks to the Author)

Revised manuscript addresses my previous concerns to an acceptable extent. Therefore, I recommend to publish the manuscript in its current form.

Reviewer #1

General comment: Luo and co-authors report the near-field study of polaritons in highly doped multilayer WSe₂ which is achieved by modifying the top layer into a monolayer of tungsten oxide. Authors also demonstrate the hyperbolic nature of these polaritons by near-field probing of patterned disc-shaped nanoresonators made by two different fabrication methods. Unfortunately, I cannot recommend this manuscript for publication in Nature Communications due to the insufficient novelty, numerous unsubstantiated claims, and instances of incorrect scientific analysis.

Our response: We believe that our response letter has clearly articulated the novelty of our work and has provided justification for the claims and analyses through this point-by-point reply. In addition, we have incorporated new experiments and analyses as suggested by the reviewer.

Comment 1: Mid-infrared hyperbolic polaritons in doped WSe₂ have been demonstrated in [Science 371, 617–620 (2021)], which is not even cited by the authors. Without detailed comparison, I do not see how the demonstrated polaritons are different from those demonstrated in the Science paper.

Our response: We agree with the reviewer that it is crucial to clearly distinguish our findings from the work by Sternbach *et al.* (Science 371, 617–620 (2021)). Although their study also involved WSe₂, the polaritons they reported are fundamentally different from the ISPs investigated here. First, Sternbach *et al.* used 300–500 nm thick bulk WSe₂, which has an A exciton energy of 1.69 eV, where the optical transition occurs between the valence and conduction bands. Second, they employed a pump at 1.5 eV (12,180 cm⁻¹) to generate electron–hole pairs in excited states, and subsequently probed the transitions with a mid-infrared (MIR) laser at 910 cm⁻¹. In their case, the polariton arises from bound-carrier excitations, with the ultrahigh density of bound-carrier transitions giving rise to negative permittivity in the MIR regime. Third, their process is intrinsically non-steady-state, as the high carrier density can only be maintained within the duration of the pump pulse.

In contrast, our work focuses on intersubband transitions in WO_x/WSe₂ heterostructures, which arise from spin–orbit coupling within the valence band of few-layer WSe₂ (~1–10 nm). In this case, the carrier density is controlled by steady-state charge transfer between WO_x and WSe₂, rather than transient photoexcitation. Moreover, reducing the thickness of WSe₂ to a few layers causes dramatic modifications to its band structure compared to the bulk crystal. Consequently, by tuning the number of WSe₂ layers from 2 to 8, we achieve a wide tunability of the intersubband transition energy, ranging from 39 meV to 636 meV (9–153 THz). Bulk WSe₂, by contrast, has a fixed band structure and does not allow such control.

[REDACTED]

Fig.R0 Top, Band structure of bulk WSe₂ from Science 371, 617–620 (2021). Bottom, Band structure of 5L-WSe₂ at Γ point.

In summary, our work is fundamentally different from the Science paper, in terms of material structure, origin of the polariton, and excitation mechanism. We cite this literature in the revised manuscript and discussed the differences: “We note that this phenomenon is fundamentally different from recently demonstrated bound carrier excitation in bulk WSe₂ near the interband optical transition energy³⁷. In bulk WSe₂ crystals, the quantized subbands vanish due to the loss of quantum confinement and merge into continuous bands. Meanwhile, the hyperbolic behavior only appears under non-equilibrium conditions, in clear contrast to the ISPs explored in this work.”

Comment 2: The mechanism of the intersubband polaritonic excitation is not explained, thus it is not clear how the experimental results demonstrate the intersubband nature of polaritons. For example, why it is important to populate the second subband as stated in line 137? And why the incident light polarization is relevant for the transitions between the subbands as stated in line 140-141?

Our response: The photon energy we chose to excite the polariton is close to the intersubband transition energy. The transition rate for the intersubband transition from subband E_i to E_j can be expressed as

$G_{ij} = \frac{2\pi}{\hbar} \frac{E_0^2 e^2}{4m^* \omega^2} e_z^2 |\langle \phi_j | \widehat{p}_z | \phi_i \rangle|^2 \times \delta(k_{\perp,n_j} - k_{\perp,n_i}) \delta(E_j - E_i - \hbar\omega)$, where $\hbar\omega$ is the photon energy involved in the transition, δ is the Kronecker delta function that ensures energy conservation and $k_{\perp,n}$ is the 2D wavevector in the x-y plane. Therefore, optical transitions only occur when the polarization of the electromagnetic wave is perpendicular to the layers in the case of 2D materials. This is the well-known polarization selection rule.

Comment 3: The nano-FTIR absorption peak by itself does not indicate the presence of the specific intersubband transitions (at what energy?) as stated in line 150 and 154. For example, the near-field absorption peak can be associated with other intraband transitions with the same energy, varying ohmic loss in doped WSe₂ due to charge screening, changes in the near-field reflection coefficient of the top WO layer caused by varying carrier concentration, or even unknown phonons in the newly studied heterostructure.

Our response: Phonon-assisted intraband absorption can occur when phonons provide the momentum required for optical transitions within the same band. However, such phonons typically have very small energies in the terahertz range. The transition energies for different WSe₂ layer numbers match our DFT calculations very well, as illustrated in the manuscript. In addition, angle-resolved photoemission spectroscopy (ARPES) measurements on the 5L-WSe₂ sample confirm that the valence subband structure is consistent with our DFT results (Fig.R1). This agreement is not limited to the 5-layer sample—measurements on other layer numbers, such as 3L-WSe₂ and 6L-WSe₂, also yield optical response energies in close agreement with the calculated

Fig.R1 Top, Band structure of bulk WSe₂. Bottom, Band structure image of 5L-WSe₂ at Γ point measured with μ -ARPES Red dashed lines indicate the subbands.

measurements on other layer numbers, such as 3L-WSe₂ and 6L-WSe₂, also yield optical response energies in close agreement with the calculated

intersubband transition energies. This strong layer dependence excludes the possibility of random phonon-assisted processes, which are typically insensitive to layer thickness. Moreover, intersubband optical transitions and near-field measurements have also been demonstrated by Schmidt et al. (Nat. Nanotechnol. 13, 1035 (2018)).

Comment 4: *The near-field fringes analysis presented in Supplementary Note 5 seems to be incorrect. First of all, the near-field inset in S.Fig. 3 does not correspond to the fitted line profile. Furthermore, the near-field images in the main text Fig. 2 only show a single fringe, while S.Fig. 3 shows at least two fringes. Where did this data come from? But most importantly, the weak near-field signal with only 1-2 fringes cannot be correctly fitted by damped sine or cosine functions because the zero coordinate is unknown. This ambiguity introduces a very big error in the fitted parameters. The proper way to fit weak near-field fringes is to use complex near-field data as was demonstrated in [Nat. Commun. 13, 1374 (2022)]. Because of this fitting error, authors greatly overestimated the polariton quality factor, claiming $Q = 20$ which is typical for phonon-polaritons in hBN [Nature Mater. 17, 134–139 (2018)] and four times exceeds $Q \sim 5$ of graphene plasmons [Nature 487, 82–85 (2012)]. Although the fitting of polariton wavelength is less sensitive to the zero-coordinate problem, I encourage authors to verify all dispersion data extracted from near-field images by using proper fitting method.*

Our response: We apologize that the profile we fitted is flipped over the inset in Supplementary Fig.3. We have fixed this problem. The data come from the same sample with same excitation energy but optimized color contrast for the fringe's visibility. The near-field images in Fig.2 of main text did show two fringes, as we have already shown the line profiles in Fig.2d.

For the fitting in the original manuscript, we followed a similar approach as described in the literature (Nat. Commun. 13, 1374 (2022)), as noted by the reviewer. In the initial analysis, we considered only the real part of the conductivity σ along with the geometric loss factor \sqrt{x} . When two or more fringes are present, fitting using only the real part provides results nearly equivalent to a full complex-valued analysis. We have now carried out both fitting procedures. As shown in Fig. R2, the complex-valued analysis, performed following the method in the literature, yields parameters that reproduce the spiral pattern. Using the same parameter q , the line profiles are also well-fitted, essentially identical to the fits using only the real part, with differences within the experimental errors. Therefore, the choice of fitting method does not significantly affect our results. **This discussion has been added to Supplementary Note 5.**

Regarding the Q factor, the literatures mentioned by the

Fig.R2 Fitted line-profile with complex-valued analysis (top) and exponentially decaying sinusoidal function (middle). Fourier transform analysis (bottom).

reviewer used two different types of calculations. For hBN (Nature Mater. 17, 134–139 (2018)), they used Fourier analysis of the line-profile and define the figure of merit as $Q = \omega_c/HWHM$, where ω_c is the peak position in the FFT and HWHM is the half-width at the half-maximum. If we use this type of FOM, our Q is about 2 (Fig.R2 bottom). For graphene (Nature 487, 82–85 (2012)), the Q factor is defined as $Q = q'/q''$, where $q' = 2\pi/\lambda_p$ and $q'' = 1/L$. We used the later one to calculate the Q factor for ISPs. We did a more accurate propagation analysis and found $Q \sim 15.8$. $Q \sim 5$ is the value of bare graphene without doping. The quality factor of plasmon in 2D materials is related to the carrier density and mobility. When the doping level is high in the WO_x /graphene heterostructure, the Q factor of graphene plasmon can also reach ~ 16 (Nat. Mater. 22, 838–843 (2023)). On the other hand, by making the graphene flake suspended, Q factor can be further increased to 33 (Nat Commun 13, 1465 (2022)). When decreased to cryogenic temperature, the Q factor of graphene plasmon can reach up to 130 (Nature 557, 530–533 (2018)). In our study, we employed high-quality WSe_2 crystals grown by the flux method to minimize the defect density.

Comment 5: There can be no radiative loss for such a high-momentum polaritons even on Si substrate as stated in line 233.

Our response: The thickness of the WO_x/WSe_2 heterostructure is much thinner comparing to the polariton wavelength. The confinement of the polariton is not as good as the thick slabs. Therefore, the radiative loss into the substrate is significant. This has also been observed for the exciton polariton waveguide mode in WSe_2 few layers regardless of the momentum of the polaritons (Phys. Rev. B 100, 121301, (2019)). In our study we found that Au mirror can improve the propagation length of the ISP. See also the Fig. R8 bellow.

Comment 6: The quality factor definition in line 259 is invalid for near-field amplitude because it has strongly non-linear dependency on the excitation energy which is not compensated by normalization.

Our response: The quality factor we referred here is the quality factor of the resonator which can be expressed as $Q = \omega/\Delta\omega$, where ω is the resonance frequency and $\Delta\omega$ is the full width half maximum (FWHM) of the cavity response. The non-linear dependency on the excitation energy has already been considered in the absorption spectrum. It has also been used by Koppens group (Nat. Mater. 23, 499–505 (2024)) and Capasso group (Sci. Adv. 4 (6), eaat7189 (2018)) to calculate the quality factor of the cavity where phonon polaritons in h-BN is confined. We added these references in the revised manuscript.

Comment 7: Quality of figures must be improved. Some panels are too small to see the claimed features (for example, where are the green/yellow and blue/red points in Fig. 1d and e as stated in line 131-132?).

Our response: Those are the residual text in the manuscript. We have now deleted this wrong sentence. We have also checked through the manuscript for other errors and revised accordingly.

Comment 8: Contents delivery and manuscript structure must be improved. The theoretical introduction in lines 97-104 lacks definitions and feels too technical for the main text. Discussion of SNOM operation principle is missing, which is crucially important to understand nano-FTIR data and near-field images.

Our response: We added the missing definitions to the theoretical introduction part and simplified this part. We also added discussion of the SNOM operation principle as reviewer suggested. We added two

paragraphs in the revised manuscript to explain the operation principle. “An s-SNOM equipped with a broadband pulsed laser is utilized to carry out the nano-Fourier-transform infrared spectroscopy (nano-FTIR). The infrared excitation laser beam is focused on the metallized tip of the atomic force microscope (AFM), generating an enhanced optical field that interacts with the heterostructure sample beneath. With that configuration we are able to characterize the intersubband transitions in $\text{WO}_x/4\text{L-WSe}_2$ at near-field with out-of-plane excitation polarization.” And “To effectively couple intersubband transitions to free-space photons and excite ISPs in the WSe_2 quantum wells, we employed s-SNOM with both a continuous-wave quantum cascade laser and a pulsed optical parametric oscillator laser to sweep across the transition energies. This configuration introduces an out-of-plane light component that drives the intersubband transitions within the heterostructure, thereby giving rise to ISPs.”

Reviewer #2

General comment: Luo et al. present a compelling and timely study on the realization and manipulation of intersubband polaritons in van der Waals quantum wells. The innovation of this work lies in the use of a self-limiting native oxide to achieve high-density hole doping in few-layer WSe₂, which represents a significant materials engineering achievement. This technique successfully enables strong intersubband transitions, providing a robust platform for investigating mid-infrared polaritonic phenomena. The subsequent demonstration of propagating hyperbolic ISPs, their electrical tunability, and their confinement in nanoresonators constitutes a substantial advance for the field of nanophotonics. The work is well-structured, the data is generally of high quality, and the findings hold promise for the development of tunable mid-infrared devices.

While the research presents an important and timely advancement, several key aspects of the analysis and interpretation require further clarification and strengthening to fully substantiate the central claims of the manuscript. I recommend publication after the authors have addressed the following major points.

Our response: We thank the reviewer pointing out that our work presents an important and timely advancement. We have answered all the comments raised by the reviewer.

Comment 1: The observation of hyperbolic ISPs in this work is an important discovery. However, the reviewers have some questions regarding this definition. Typically, polaritons in semiconductors are exciton-polaritons, whereas the authors classify this mode as an intersubband plasmon-polariton. In addition, how do the authors distinguish this polariton mode from a waveguide mode? The reviewers hope the authors can provide further analysis and clarification regarding these questions.

Our response: In semiconductors, either exciton polaritons or plasmon polaritons can be excited depending on the excitation conditions. A representative example is black phosphorus (BP), where both exciton polaritons (Nat. Commun. 12, 5628 (2021)) and plasmon polaritons (Nat. Commun. 15, 709 (2024)) have been theoretically predicted and experimentally observed. In our case, the A-exciton absorption energy is 1.74 eV for monolayer and 1.69 eV for bulk WSe₂. Here, we employed mid-IR excitation in the 0.1–0.2 eV range, which is far below the excitonic bandgap and thus cannot generate excitons. Instead, the measured near-field optical response energies match the intersubband transition energies for different layer thicknesses, in good agreement with the theoretical prediction in Ref. 22.

In general, when the material is not polarized strongly, i.e., permittivity tensors having no negative term, the polariton mode is called waveguide mode. Therefore, the waveguide mode typically has a small wavelength compression ratio λ_0/λ_p , for example exciton polariton waveguide mode in MoSe₂ with $\lambda_0/\lambda_p < 2$ (Nat. Photon. 11 356 (2017)). Also, the slab of the sample is usually thick to support the coupling between the free-space photon to the waveguide. In our case, the permittivity along the z-axis is negative due to the strong carrier doping. We observed large wavelength compression ratio up to $\lambda_0/\lambda_p = 27$, comparable to surface plasmon in graphene or phonon polariton in h-BN. Meanwhile, the thickness of our heterostructure is down to 3 nm.

Comment 2: The origin of the transition is central to the paper, but its depiction in Figure 1 could be improved. As presented, it is difficult to visually identify the specific subbands involved in the measured mid-IR transition from the DFT calculations (Fig. 1d-e). We strongly suggest adding graphical indicators

(e.g., arrows) directly onto the band structure in Figure 1d to illustrate the transition between the second and third valence subbands at the Γ point.

Our response: We thank the reviewer to provide such constructive advice. We have revised the Fig.1 accordingly. The red arrow is added to the Fig.1d and 1e.

Comment 3: The claim that the 'hot-ring' feature in nanodisks (Fig. 3) is conclusive proof of conical hyperbolic ray propagation requires more rigorous evidence, as similar ring-like features can be produced even in isotropic resonators (Adv. Funct. Mater. 2019, 29, 1904662).

Our response: The ring-like features reviewer mentioned in Adv. Funct. Mater. 2019, 29, 1904662 are standing waves formed in the resonators as we did in Fig.4 (see also Sci. Adv. 4, eaat7189 (2018)). In contrast, in order to clarify the conical ray propagation, we redesign the experiment by using the Au disk under the WO_x/WSe_2 heterostructure. Simulation result is shown in Fig.R3. The ray propagating both inside and outside of the Au disk region creates two hot-ring. While as we shown in Fig.4 of the main text, the resonator will confine the polariton propagation inside the resonator and no hot-ring outside of the resonator will be created. This method is commonly used as a proof of conical hyperbolic ray propagation as demonstrated in Ref.41 and 42.

Fig.R3 FDTD simulated field profile for the WO_x/WSe_2 heterostructure

Comment 4: The reliance on permittivity values from s-SNOM fitting warrants a more critical discussion, as extracted values can diverge from the actual dielectric function (Nature Reviews Materials 2025, 26). We suggest the authors justify their fitting procedure and validate the results with other measurements, such as ellipsometry or far-field optical characterization.

Our response: First, nano-FTIR is well suited for probing the out-of-plane permittivity because the enhanced tip-sample interaction is sensitive to the out-of-plane electromagnetic field. In contrast, IR ellipsometry is challenging in our case due to the small sample size, and far-field FTIR is not highly sensitive to out-of-plane absorption since the incident light is predominantly polarized in-plane. Nevertheless, our results are supported by DFT calculations of the dielectric function. As shown in Fig. R4, the imaginary part of the permittivity (ϵ_2) is nearly zero for pristine 5L- WSe_2 , whereas doping with WO_x introduces an absorption peak around 146 meV. This feature is in excellent agreement with both our nano-FTIR measurements and the extracted ISP dispersion.

Fig.R4 DFT calculated imaginary part of the permittivity for 5L- WSe_2 (top) and $\text{WO}_x/4\text{L-WSe}_2$ (bottom).

Comment 5: The real-space images in Figures 2 and 3 are primary evidence for polariton propagation, but the interference fringes have limited visibility. This may introduce uncertainty in the extracted wavelength. To strengthen the evidence, could the authors provide additional images with improved signal-to-noise ratio?

Our response: We have now adopted the Au mirror method as introduced by Chen et al. (Nat. Mater. 22, 860–866 (2023)). As shown in Fig.8, we can now observe at least three fringes which minimized the uncertainty dramatically.

Comment 6: The reviewers find the variation in signal intensity with disk size, as observed in Figure 4b, to be a very interesting phenomenon. However, the current explanation for this observation is too brief. The reviewers request that the authors provide a more detailed and thorough explanation.

Our response: We agree with the reviewer that the observation of the disk size dependent intensity is of interest. The rest part of the Fig.4 is actually the explanation why such dependence existed. Still, we provide more details and explanation in the revised manuscript: “The propagation of the ISP in the nanoresonators can be described by the wave equation $\rho(i\omega)^2 = \nabla^2 \rho v_p^2(\omega)$, where ρ denotes the integrated two-dimensional charge density and v_p is the group velocity⁴⁸. For a disk-shaped resonator, the solution of ρ can be expressed in terms of Bessel functions, with the eigenvalues k_{sn} determined by the disk diameter. Consequently, because the ISP can resonate only with resonators of specific sizes at a given excitation photon energy, tuning the E_{ph} enables the systematic mapping of the relationship between disk diameter and the ISP resonant modes (Fig.4b).”

Comment 7: The manuscript would benefit from a more thorough discussion of its relationship to existing literature. Specifically, the use of WO_x for charge-transfer doping to tune polaritons has been demonstrated in graphene plasmonic systems (Nature Materials 22, 838–843, 2023). The authors should explicitly discuss this work and further clarify the novelty of their WSe_2 -based intersubband polariton platform. Furthermore, to properly contextualize their findings, the authors should incorporate citations to other recent key developments in hyperbolic polaritons, such as studies on topological transitions in tunable materials like graphene-based heterostructure, demonstrations of negative refraction, and the discovery of hyperbolic plasmons in the visible range in novel bad metals.

Our response: We did cite the work mentioned by the reviewer as Ref.40 in our manuscript. Our work is quite different from that literature. Brian et. al demonstrated using WO_x layer as a doping method to engineer the plasmon in graphene. Our work features the first observation of the ISP in highly doped few-layer WSe_2 which forms natural quantum wells. As reviewer suggested, we add comparison sentences in the introduction as “This approach has been demonstrated to efficiently modulate the carrier density in graphene and thereby engineer plasmon polaritons³¹. Although promising for carrier density control, its application to the study of ISPs remains elusive.” We also add more thorough discussion of the hyperbolic polaritons in the revised manuscript. “The layer-dependent hyperbolicity highlights ISPs as a promising addition to the family of hyperbolic polaritons, complementing other recently demonstrated phenomena such as topological transitions⁴², negative refraction^{43,44}, and low-loss propagation⁴⁵.”

Minor comment 1: Regarding the use of italics on lines 113 and 124, the intended meaning or emphasis is unclear to the reviewers. Please clarify the purpose of using italics in these instances.

Our response: The italics on lines 113 and 124 are for emphasis. We have now removed them.

Minor comment 2: The meaning of the dashed line in Figure 1a is not explained in the caption. Please add a definition.

Our response: Dashed lines in Figure 1a illustrate the subband energy positions. We have added it to the figure caption.

Minor comment 3: There is an inconsistency in the formatting of the variable for the number of layers, N . For example, it appears non-italicized on line 314 but italicized on line 312. Please ensure uniform formatting throughout the manuscript.

Our response: We have corrected this inconsistency and corrected all the fonts issue.

Minor comment 4: The method used to determine the error bars in all figures should be explicitly stated.

Our response: The errors came from the determination of the polariton wavelength from the line profile. We determine the error range by fitting the fringe periodicity.

Minor comment 5: The black dashed line is missing in Figure 2d.

Our response: We have added the missing black dashed line to Figure 2d.

Reviewer #3

General comment: *The manuscript authored by Yue Luo presents a study on the observation and manipulation of hyperbolic intersubband polaritons (ISPs) in WO_x/WSe_2 heterostructures, focusing on a heterostructure based on oxidized tungsten diselenide ($WO_x/4L-WSe_2$). The topic is of significant interest, as ISPs have long been theoretically proposed but remained experimentally elusive. The oxidation process induced by the top layer of the heterostructure enhances the doping levels, leading to intersubband transitions with sufficient oscillator strength and enabling the system to support ISP formation. This approach is well supported by DFT calculations, including detailed studies of the influence of each WSe_2 layer.*

The work further demonstrates real-space propagation of these ISP polaritons using s -SNOM (scattering-type scanning near-field optical microscopy). While the results are impressive, the polaritonic patterns are not particularly compelling, as only a single fringe is observed in the experiments. The authors should emphasise the advantages of this configuration compared to other polaritonic materials, such as hBN or MoO_3 (which offer long lifetimes) or graphene (which allows for stronger electrical tunability). In addition, the authors design and characterize polaritonic nanoresonators with varying diameters. The results are well presented and supported by numerical calculations. As a final remark, it would be valuable to explore the combination of electrical tunability with nanoresonators to investigate whether doping can enhance the resonances.

The manuscript addresses an exciting and novel experimental approach to effectively couple intersubband transitions to free-space photons. However, to reach the standards of Nature Communications, several conceptual and technical points require clarification or further development. I encourage the authors to revise the manuscript accordingly, addressing the following points.

Our response: We thank the reviewer by point out that our work addressed an exciting and novel experimental approach. We also thank the reviewer by providing constructive comments. We have addressed the points in detail.

Comment 1: *In Fig. 2, the authors attribute ISP excitation solely to tip-launched polaritons, without accounting for potential edge contributions. Since edge-launched polaritons often dominate and only a single fringe is observed in the near-field experiment, the contribution from the edge should be considered, or the authors should justify their assumption.*

Our response: Indeed, as the reviewer correctly pointed out, edge-launched polaritons dominate when the probe is positioned near the edge. We agree and have considered the ISP observed in our near-field nano-imaging results to be edge-launched polaritons. As stated in lines 178–180, the bright–dark fringe spacing corresponds to half of the polariton wavelength ($\lambda_{ISP}/2$). In contrast, for tip-launched polaritons, the fringe period is equal to the full polariton wavelength (λ_{ISP}).

Comment 2: *The permittivity values of WSe_2 with high carrier density under charge transfer (Fig. 2a) suggest that out-of-plane ray propagation of polaritons might occur, analogously to hBN or MoO_3 flakes. However, in Fig. 2e, the experimental wavelengths are compared with the fundamental polariton mode. This mode's wavelength can differ significantly from that of the out-of-plane ray, which is determined solely by the heterostructure thickness and permittivity. Therefore, the authors should verify the type of*

mode being probed in their *s*-SNOM data. In this sense, numerical simulations (similar to those in Fig. 4g) would be helpful.

Our response: The experimental data in Fig.2e are extracted from the line profile of the fringes features in the nanoimaging results. The momentum k is calculated as $k = 2\pi/\lambda_{\text{ISP}}$. In such way, they are compared with the fundamental polariton mode. In contrast, for the out-of-plane ray propagation, we plotted ΔR vs. photon energies in Fig.3f. We did probe both modes in the different experiments.

Fig.R5 Simulated field profiles for the WO_x/WSe_2 heterostructure at different excitation energies that show the fundamental mode.

As the reviewer suggested, we did numerical simulation similar to Fig.4g in the main text to investigate the polariton mode in the heterostructure (Fig.R5). The simulation results suggested that the polariton can propagate along the heterostructure within the spectral range. Also, the polariton wavelength is inversely proportional to the excitation wavelength.

Comment 3: The extracted wavelengths for 144 meV and 149 meV are 317 nm and 453 nm (lines 182-183), respectively. This would imply corresponding momenta of approximately $2 \times 10^7 \text{ m}^{-1}$ and $1.4 \times 10^7 \text{ m}^{-1}$. However, these values appear inconsistent with the data shown in Fig. 2e, and the discrepancy should be addressed.

Our response: We identified an error in plotting Fig. 2e, where an additional factor of two was mistakenly applied (considering the edge-launched polariton), although this factor had already been accounted for in the polariton wavevector. This has been corrected, and the corresponding fitting has been updated accordingly (Fig.R6).

Fig.R6 Revised dispersion relation.

Comment 4: A key point emphasized by the authors is the electrical tunability of the ISP wavelength. While this is a valuable feature, prior studies in dopable materials supporting polaritons or even in heterostructures of MoO_3 or *h*BN combined with graphene, have achieved greater tunability while maintaining higher propagation lengths. Lines 275–277 (among others) should be revised for clarity and technical precision.

Our response: The key advantage of our platform is that it enables tunability without the need for introducing an additional material to form hybridized polaritons. In contrast, even with graphene hybridization, the achievable tunability is limited to $\sim 15\%$ (Nat. Nanotechnol. 10, 682–686 (2015)). Here, we primarily provide a proof of concept that intersubband (ISB) transitions in highly doped TMD multilayers can be electrically tuned, offering a promising route toward tunable photonic devices.

Comment 5: In Fig. 2f, the images and field profiles are absent for the labelled points, and no such data is found in the Supplementary Material either. Clarifying whether these are experimental points or simulations would be necessary.

Our response: The data points in Fig.2f are experimental results extracted from the line profiles taken at different gate voltages. Figure R7 shows the line profiles we used to extract the polariton momentum. **We have now included the line profiles in the supplementary material.**

Comment 6: In Fig. 2g, nearly a single fringe is observed for the $WO_x/3L-WSe_2$ and $WO_x/5L-WSe_2$ cases, making it difficult to determine the presence of ISP. Moreover, the location of the flake boundary is not clearly indicated. Line profiles should be included to support the appearance of ISP in these heterostructures.

Our response: We displayed line profiles in Fig.2d. The indication of the location of the flake boundary (black dashed line in Fig.2d) was missing. We have now revised figure. In addition, to observe more fringes, we adapted the image polariton method by adding Au mirror with dielectric layer between the sample and Au mirror. We can now observe at least three fringes, which is helpful for analyzing the ISP (Fig.R8).

Fig.R7 Extracted line profiles at different gate voltages.

Fig.R8 a) Nano-imaging of the $WO_x/5L-WSe_2$ heterostructure on the Au mirror at various excitation energies. Red dashed line indicates the flake boundary. b) Line-profiles extracted at each excitation energies. c) Experimental dispersion relation calculated from the fringe periodicity.

Comment 7: In Fig. 3, the authors state that ISP polaritons are launched at the cavity edge ($WO_x/4LWSe_2$) and propagate into the surrounding $5L-WSe_2$. This seems contradictory, as $5L-WSe_2$ alone does not support ISP polaritons. If ISP modes are not sustained in $5L-WSe_2$, then observed rings cannot be attributed to such polaritons. The origin and nature of the rings appearing in the cavity surroundings must be clarified.

Our response: We agree with the reviewer that the hot-ring features in our embedded nanodisc experiment is not solely due to the directional propagation. Therefore, we did the well-accepted Au disk experiment to prove the directional ray propagation (Fig.R9). **We have now replaced the Fig.3 in the main text.**

Fig.R9 Hyperbolic ISP revealed by nano-imaging on Au nanodisk. **a**, Schematic of hyperbolic polaritons launched at the edge of the Au nanodisk, which travel along conical trajectories and form a bright ring. The distance between the ring and the edge of the Au nanodisk is ΔR . **b** and **c**, Topography (**b**) and near-field scattering amplitude $s(\omega)$ image of $\text{WO}_x/4\text{L-WSe}_2$ plasmonic cavity with diameter $D = 300$ nm and excitation energy $E_{\text{ph}} = 145$ meV (**c**). Scale bar: 300 nm. **d**, Line profile of the measured near-field amplitude taken at the dashed line in **c**. **e** and **f**, Near-field scattering amplitude $s(\omega)$ image of sample on the Au nanodisk that is partially covered by $\text{WO}_x/4\text{L-WSe}_2$ and $\text{WO}_x/6\text{L-WSe}_2$ simultaneously and excited with $E_{\text{ph}} = 145$ meV (**e**) and $E_{\text{ph}} = 146$ meV (**f**), respectively. Red dashed lines indicate the boundary. Scale bar: 150 nm.

Comment 8: It remains unclear why the authors employ a fundamental mode interpretation in Fig. 2e, but refer to an out-of-plane ray model in Fig. 3 (lines 216–218). A clear justification of the applied framework should be provided to properly interpret the different figures.

Our response: Lines 216–218 refer to a separate experiment designed to probe the out-of-plane ray mode (Fig. 3a). In this case, polaritons are launched from the edge of the structure and propagate at an angle given by $\arctan(i\sqrt{\epsilon_{xy}}/\sqrt{\epsilon_z})$. In contrast, Fig. 2e presents experimental data extracted from nano-imaging measurements, where the interference of edge-launched polaritons produces bright–dark fringes. These are two distinct experiments serving different purposes: the out-of-plane ray propagation demonstrates the hyperbolic nature of the mode, whereas the near-edge interference fringes reveal the polariton dynamics.

Comment 9: The manuscript lacks discussion of the lifetimes of the observed ISP polaritons. Considering the short propagation length, this is an important parameter to assess the practical viability of the system and should be included.

Our response: The lifetime of the ISP polariton can be calculated using $\tau_{\text{ISP}} = L/v_g$, where L is the propagation length and v_g is the group velocity. The propagation length is extracted from the fitting parameter of the line profile while v_g is calculated using the dispersion relation $v_g = \frac{1}{\hbar} \frac{dE}{dk}$. Figure R10 shows

v_g as a function of E , where we found $v_g = 1.71 \times 10^6$ m/s or $0.0057c$ at $E = 0.144$ eV. Therefore, the lifetime is

$$\tau_{\text{ISP}} = \frac{L}{v_g} = \frac{0.8 \times 10^{-6}}{1.71 \times 10^6} = 0.468 \text{ ps} \quad \text{for}$$

Fig. R10. Group velocities for WO_x/4L-WSe₂ (a) and WO_x/5L-WSe₂

WO_x/4L-WSe₂ on the SiO₂/Si substrate. When using the Au substrate, $\tau_{\text{ISP}} = \frac{1.3 \times 10^{-6}}{5 \times 10^5} = 2.6 \text{ ps}$ for WO_x/5L-WSe₂. This is comparable to the lifetime of MoO₃ (Nature 562, 557–562 (2018)) and longer than graphene plasmon polariton measured at cryogenic temperature (Nature 557, 530–533 (2018)). We have now provided this number in the revised manuscript.

Comment 10: In line 229, the sentence "Although the nano-imprinted nanodiscs can be used as a demonstration of the directional propagation..." is ambiguous. In what precise sense is directionality demonstrated? Please elaborate.

Our response: We have already replaced this part with the Au disk experiment.

Comment 11: In Fig. 1d and Fig. 1e, it is difficult to distinguish the green, yellow, and red markers.

Our response: We do not have yellow and red markers in Fig. 1d and Fig. 1e. The colors in the DFT calculated band structures only help to identify the contribution from WSe₂ and WO_x.

Comment 12: Several supplementary note references appear incorrectly referenced, e.g., lines 154 and 184.

Our response: We have corrected the referencing in the manuscript.

Comment 13: The thicknesses of the individual layers in Fig. 2 should be clearly labelled.

Our response: A layer of WSe₂ is about 0.7 nm, we labelled the layer numbers through the manuscript.

Comment 14: The assumption of in-plane isotropy is only specified in Fig. 4 (lines 247–248). This assumption should be stated earlier in the manuscript.

Our response: We now state this assumption earlier when we are discussing the in-plane permittivity.

Reviewer 1

Comment 1: The agreement between the optical responses (nano-FTIR peaks and ISP momenta) and the calculated intersubband transition energies is still missing. Current Figures 1d,e and 2e, and Supplementary Figure 7 do not show subbands responsible for the experimental data at given energy – the y-axis limit of every band diagram is too large to resolve ~ 100 meV excitation energies. Please show the bands responsible for the corresponding optical responses observed in the experiments (similar to the lower panel in Figure R0 with clearly marked intersubband transition energies).

Our response: We have revised the manuscript to mark the transition energies in the figures. As shown in Fig. R1, we marked the transition energies for the top three subbands in the VB. We have also added this to the Supplementary Note 3. For Figure 1d and e, we mentioned the transition energy in the figure caption. We also revised Supplementary Figure 7 with clearly marked intersubband transition energies.

Fig.R1. DFT calculated band structures for 5L-WSe2 and WO_x/4L-WSe2 heterostructure.

Comment 2: Please numerate equations for the intersubband transition energies in the first paragraph of the Results section (lines 104 and 105) and refer to them in the text where needed for different cases. For example, Figure 1b shows the DFT and analytical calculations for the first and second subbands, but Figure 1f illustrates the resonance between the incident photon and the intersubband transition between the second and third subbands as mentioned in main text line 142. This is confusing. Are these transitions equally probable and have equal energies? If not, they should be discriminated. Then, when discussing the experiments, authors refer to “intersubband transition energy” several times without indicating which transition is assumed. This needs to be clarified.

Our response: The transitions between E_1 and E_2 as well as E_2 and E_3 are both accessible with different intersubband transition energies. Figure 1b shows the calculation from first and second subbands of pristine WSe₂ without WO_x layer involved. However, as we mentioned in the main text and Supplementary Note 3, in the WO_x doped case, the first subband is originated from the WO_x layer. Therefore, we only consider the $E_{2,o} \rightarrow E_{3,o}$ intersubband transition in the WO_x/WSe₂ heterostructure which has the similar intersubband transition energy of the $E_{1,p} \rightarrow E_{2,p}$ in the pristine layer. The intersubband transition energies we mentioned are consistent between the DFT calculation and the experiments. We numerated the equations as reviewer suggested and we marked the transition bands in the revised manuscript.

Comment 3: Results of the near-field ISP fringes analysis still require revision. First of all, $s(x)$ in Supplementary Figure 5 (which, I believe, was used to calculate Q) shows three near-field fringes that are never observed in any near-field image in the original manuscript. The inset does not correspond to $s(x)$ either. I plotted the two $s(x)$ profiles of the inset image and the results are not even close to $s(x)$ in Figure S5: Please replace the inset with the near-field image used for the $s(x)$ analysis. Even better than that, please use the newly obtained near-field images on gold mirror (Supplementary Figure 9) to calculate Q . This data is of better quality and should provide much more accurate value of Q .

Additionally, authors are not correct regarding the definition of Q in the paper by Giles [Nature Mater. 17, 134–139 (2018)]. As Giles stated on page 137: “A relevant figure of merit (FOM) may be defined simply as $Q = \text{Re}(q)/\text{Im}(q)$ ” – the same Q used by the authors. According to Fig. 5, measured and calculated Q of PhP in naturally abundant hBN is ≈ 15 , which corresponds to 5-6 clearly visible near-field fringes associated with the tip-launched mode (Fig. 3). The authors may also refer to the work by Chen [Nature Mater. 22, 860–866 (2023)] where polaritons in Ag₂Te are imaged by s-SNOM and produce almost two near-field fringes over gold mirror (Fig. 2), corresponding to $Q \sim 2$ and lifetime of ~ 0.2 ps. Finally, regarding the graphene plasmons reported by Basov’s group in [Nature 487, 82–85 (2012)], plasmons with $Q \approx 5$ in naturally doped graphene at zero gate bias produced 3-4 clearly visible near-field interference fringes formed by the tip-launched plasmons near the graphene edge (Fig. 1b). These fringes are very similar to the fringes observed by the authors in the heterostructures on gold mirror, suggesting $Q \sim 5$. Since $Q/2\pi$ is equal to the normalized propagation length measured in polariton’s wavelength regardless of the physical nature of polaritonic wave or specific material properties, I see no evidence for high $Q > 10$ in the presented data.

Our response: We agree with the reviewer that the accuracy of fitting using data with only two fringes is limited. To improve the signal quality, we averaged over 10 lines, and we have updated the inset with enhanced contrast. We also agree that the data acquired on the gold mirror exhibits a better signal-to-noise ratio, where we observed a maximum Q of 16.9. To avoid ambiguity, we have removed the quality factor statement from the main text and now report the Q value for the gold mirror case in Supplementary Note 9.

Fig.R2 Line profile with better contrast inset.

Comment 4: Please specify thickness of the flakes shown in Figure 2.

Our response: The flakes shown in Figure 2 are WO_x/4L-WSe₂ which has the thickness of ~ 4.5 nm (See also Supplementary Figure 1c). We have mentioned the thickness in the revised Figure 2.

Reviewer 2

Comment 1: In the captions of Figures 2e, 2f, and 3f, it would be helpful to include an explanation of the origin and calculation method of the error bars.

Our response: We have now added the explanation of the error bars calculation in the revised manuscript as “The error bars are determined from the fitting uncertainties of the fringe periodicities.”

Comment 2: In Figure 3f, the alignment of the panel label “f” could be adjusted to match the order and style of the other figure labels.

Our response: The figure has been revised to match the order and style.

Comment 3: In Figures 3b and 3c, the boundaries of the structures are currently depicted using a combination of white solid lines and colored dashed lines, which may slightly interfere with the presentation of the data.

Our response: We have revised the Figure 3 accordingly.

Comment 4: In Figures 3b–3e, the colors of the marker lines and the data lines do not correspond optimally. For my understanding, the purple dashed line in Figure 3c might be better represented as a dark blue dashed line to improve consistency with the other panels.

Our response: We have changed the entire Figure 3. There issue has been resolved.

Reviewer 3

Comment 1: The authors demonstrate that the gold substrate enhances ISP propagation, leading to longer polariton propagation lengths and lifetimes. Such extended lifetimes (2.6 ps), comparable to those reported for hBN or MoO₃, represent a significant result for practical applications, such as the structure presented in Figure 4. I encourage the authors to emphasize this point more explicitly, for instance by reporting the lifetime value of this structure either in the main text or in Supplementary Note 9.

Our response: We have mentioned this value in Supplementary Note 10.

Comment 2: In Fig.3 the expression $\Delta R/d = |\tan\theta| = i\sqrt{\epsilon_{xy}\epsilon_z}$ is employed in the analysis. The authors should compare the experimentally extracted value with the one obtained from substituting the permittivity values.

Our response: We tried to compare the experimentally extracted value with the calculated one. Although the trend is correct, the $\Delta R/d$ is off from the theoretical data, as shown in Fig.R3. This could be caused by the tilting when the very thin layer covered over the quite thick Au disk.

Fig.R3. ΔR as a function of polariton energies.

Review of “*Observation of hyperbolic intersubband polaritons in native-dielectric-doped van der Waals semiconductor quantum wells*”.

The manuscript authored by Yue Luo presents a study on the observation and manipulation of hyperbolic intersubband polaritons (ISPs) in WO_x/WSe_2 heterostructures, focusing on a heterostructure based on oxidized tungsten diselenide ($WO_x/4L-WSe_2$). The topic is of significant interest, as ISPs have long been theoretically proposed but remained experimentally elusive. The oxidation process induced by the top layer of the heterostructure enhances the doping levels, leading to intersubband transitions with sufficient oscillator strength and enabling the system to support ISP formation. This approach is well supported by DFT calculations, including detailed studies of the influence of each WSe_2 layer.

The work further demonstrates real-space propagation of these ISP polaritons using s-SNOM (scattering-type scanning near-field optical microscopy). While the results are impressive, the polaritonic patterns are not particularly compelling, as only a single fringe is observed in the experiments. The authors should emphasise the advantages of this configuration compared to other polaritonic materials, such as hBN or MoO_3 (which offer long lifetimes) or graphene (which allows for stronger electrical tunability). In addition, the authors design and characterize polaritonic nanoresonators with varying diameters. The results are well presented and supported by numerical calculations. As a final remark, it would be valuable to explore the combination of electrical tunability with nanoresonators to investigate whether doping can enhance the resonances.

The manuscript addresses an exciting and novel experimental approach to effectively couple intersubband transitions to free-space photons. However, to reach the standards of Nature Communications, several conceptual and technical points require clarification or further development. I encourage the authors to revise the manuscript accordingly, addressing the following points:

1) In Fig. 2, the authors attribute ISP excitation solely to tip-launched polaritons, without accounting for potential edge contributions. Since edge-launched polaritons often dominate and only a single fringe is observed in the near-field experiment, the contribution from the edge should be considered, or the authors should justify their assumption.

2) The permittivity values of WSe_2 with high carrier density under charge transfer (Fig. 2a) suggest that out-of-plane ray propagation of polaritons might occur, analogously to hBN or MoO_3 flakes. However, in Fig. 2e, the experimental wavelengths are compared with the fundamental polariton mode. This mode's wavelength can differ significantly from that of the out-of-plane ray, which is determined solely by the heterostructure thickness and permittivity. Therefore, the authors should verify the type of mode being probed in their s-SNOM data. In this sense, numerical simulations (similar to those in Fig. 4g) would be helpful.

3) The extracted wavelengths for 144 meV and 149 meV are 317 nm and 453 nm (lines 182-183), respectively. This would imply corresponding momenta of approximately $2 \times 10^7 \text{ m}^{-1}$ and $1.4 \times 10^7 \text{ m}^{-1}$. However, these values appear inconsistent with the data shown in Fig. 2e, and the discrepancy should be addressed.

4) A key point emphasized by the authors is the electrical tunability of the ISP wavelength. While this is a valuable feature, prior studies in dopable materials supporting polaritons or

even in heterostructures of MoO₃ or hBN combined with graphene, have achieved greater tunability while maintaining higher propagation lengths. Lines 275–277 (among others) should be revised for clarity and technical precision.

5) In Fig. 2f, the images and field profiles are absent for the labelled points, and no such data is found in the Supplementary Material either. Clarifying whether these are experimental points or simulations would be necessary.

6) In Fig. 2g, nearly a single fringe is observed for the WO_x/3L-WSe₂ and WO_x/5L-WSe₂ cases, making it difficult to determine the presence of ISP. Moreover, the location of the flake boundary is not clearly indicated. Line profiles should be included to support the appearance of ISP in these heterostructures.

7) In Fig. 3, the authors state that ISP polaritons are launched at the cavity edge (WO_x/4L-WSe₂) and propagate into the surrounding 5L-WSe₂. This seems contradictory, as 5L-WSe₂ alone does not support ISP polaritons. If ISP modes are not sustained in 5L-WSe₂, then observed rings cannot be attributed to such polaritons. The origin and nature of the rings appearing in the cavity surroundings must be clarified.

8) It remains unclear why the authors employ a fundamental mode interpretation in Fig. 2e, but refer to an out-of-plane ray model in Fig. 3 (lines 216–218). A clear justification of the applied framework should be provided to properly interpret the different figures.

9) The manuscript lacks discussion of the lifetimes of the observed ISP polaritons. Considering the short propagation length, this is an important parameter to assess the practical viability of the system and should be included.

10) In line 229, the sentence "Although the nano-imprinted nanodiscs can be used as a demonstration of the directional propagation..." is ambiguous. In what precise sense is directionality demonstrated? Please elaborate.

Minor Comments:

11) In Fig. 1d and Fig. 1e, it is difficult to distinguish the green, yellow, and red markers.

12) Several supplementary note references appear incorrectly referenced, e.g., lines 154 and 184.

13) The thicknesses of the individual layers in Fig. 2 should be clearly labelled.

14) The assumption of in-plane isotropy is only specified in Fig. 4 (lines 247–248). This assumption should be stated earlier in the manuscript.

I appreciate the authors efforts to improve the quality of the manuscript by addressing many of the reviewers' comments. While improvements have been made, further revision is necessary to meet the high standard of Nature Communications. Particularly, the following critically important issues must be resolved before manuscript can be considered for publication.

1. The agreement between the optical responses (nano-FTIR peaks and ISP momenta) and the calculated intersubband transition energies is still missing. Current Figures 1d,e and 2e, and Supplementary Figure 7 do not show subbands responsible for the experimental data at given energy – the y-axis limit of every band diagram is too large to resolve ~ 100 meV excitation energies. Please show the bands responsible for the corresponding optical responses observed in the experiments (similar to the lower panel in Figure R0 with clearly marked intersubband transition energies).

2. Please numerate equations for the intersubband transition energies in the first paragraph of the Results section (lines 104 and 105) and refer to them in the text where needed for different cases. For example, Figure 1b shows the DFT and analytical calculations for the first and second subbands, but Figure 1f illustrates the resonance between the incident photon and the intersubband transition between the second and third subbands as mentioned in main text line 142. This is confusing. Are these transitions equally probable and have equal energies? If not, they should be discriminated. Then, when discussing the experiments, authors refer to “intersubband transition energy” several times without indicating which transition is assumed. This needs to be clarified.

3. Results of the near-field ISP fringes analysis still require revision. First of all, $s(x)$ in Supplementary Figure 5 (which, I believe, was used to calculate Q) shows three near-field fringes that are never observed in any near-field image in the original manuscript. The inset does not correspond to $s(x)$ either. I plotted the two $s(x)$ profiles of the inset image and the results are not even close to $s(x)$ in Figure S5:

Please replace the inset with the near-field image used for the $s(x)$ analysis.

Even better than that, please use the newly obtained near-field images on gold mirror (Supplementary Figure 9) to calculate Q . This data is of better quality and should provide much more accurate value of Q .

Additionally, authors are not correct regarding the definition of Q in the paper by Giles [Nature Mater. 17, 134–139 (2018)]. As Giles stated on page 137: “A relevant figure of

merit (FOM) may be defined simply as $Q = \text{Re}(q)/\text{Im}(q)$ – the same Q used by the authors. According to Fig. 5, measured and calculated Q of PhP in naturally abundant hBN is ≈ 15 , which corresponds to 5-6 clearly visible near-field fringes associated with the tip-launched mode (Fig. 3).

The authors may also refer to the work by Chen [Nature Mater. 22, 860–866 (2023)] where polaritons in Ag₂Te are imaged by s-SNOM and produce almost two near-field fringes over gold mirror (Fig. 2), corresponding to $Q \sim 2$ and lifetime of ~ 0.2 ps.

Finally, regarding the graphene plasmons reported by Basov’s group in [Nature 487, 82–85 (2012)], plasmons with $Q \approx 5$ in naturally doped graphene at zero gate bias produced 3-4 clearly visible near-field interference fringes formed by the tip-launched plasmons near the graphene edge (Fig. 1b). These fringes are very similar to the fringes observed by the authors in the heterostructures on gold mirror, suggesting $Q \sim 5$.

Since $Q/2\pi$ is equal to the normalized propagation length measured in polariton’s wavelength regardless of the physical nature of polaritonic wave or specific material properties, I see no evidence for high $Q > 10$ in the presented data.

4. Please specify thickness of the flakes shown in Figure 2.